# A simple parametrization of mélange buttressing for calving glaciers

Tanja Schlemm[1,2] and Anders Levermann[1,2,3]

[1]Potsdam Institute for Climate Impact Research, Potsdam, Germany
[2]Institute of Physics and Astronomy, University of Potsdam, Potsdam, Germany
[3]Lamont-Doherty Earth Observatory, Columbia University, New York, USA

**Correspondence:** anders.levermann@pik-potsdam.de

**Abstract.** Both ice sheets on Greenland and Antarctica are discharging ice into the ocean. In many regions along the coast of the ice sheets, the icebergs calve into a bay. If the addition of icebergs through calving is faster than their transport out of the embayment, the icebergs will be frozen into a mélange with surrounding sea ice in winter. In this case, the buttressing effect of the ice mélange can be considerably stronger than any buttressing by mere sea ice would be. This in turn stabilizes the glacier
terminus and leads to a reduction in calving rates. Here we propose a simple parametrization of ice mélange buttressing which leads to an upper bound on calving rates and can be used in numerical and analytical modeling.

## 1 Introduction

Ice sheets gain mass by snowfall and freezing of sea water and lose mass through calving of icebergs and melting at the surface
and the bed. Currently the ice sheets on Antarctica and Greenland have a net mass loss and contribute increasingly to sea level rise (Rignot et al., 2014; Shepherd et al., 2018b; WCRP Global Sea Level Budget Group, 2018; Eric Rignot, 2019; Mouginot et al., 2019). The ice sheet's future mass loss is important for sea level projections (Church et al., 2013; Ritz et al., 2015; Golledge et al., 2015; DeConto and Pollard, 2016; Mengel et al., 2016; Kopp et al., 2017; Slangen et al., 2017; Golledge et al., 2019; Levermann et al., 2020).

For the Greenland ice sheet, calving accounted for two-thirds of the ice loss between 2000 and 2005, while the rest was lost due to enhanced surface melting (Rignot and Kanagaratnam, 2006). Because surface melt increased faster than glacier speed, calving was responsible for a third of the mass loss of the Greenland ice sheet between 2009 and 2012 (Enderlin et al., 2014). In the future, enhanced warming (Franco et al., 2013) and the melt elevation feedback (Weertman, 1961; Levermann and Winkelmann, 2016) will further increase surface melt but also intensify the flow of ice into the ocean. Calving accounts for roughly
half the ice loss of the Antarctic ice shelves, the rest is lost by basal melt (Depoorter et al., 2013).

It is clear that calving plays an important role in past and present ice loss and is therefore very likely to play an important role for future ice loss. However, by just calving off icebergs into the ocean and considering them eliminated from the stress field

of the ice-sheet-ice-shelf system, most studies neglect the buttressing effect of a possible ice mélange, which can form within the embayment into which the glacier is calving. This study provides a simple parametrization that accounts for the buttressing effect of ice mélange on calving on a large spatial scale and that can be used for continental scale ice sheet modeling. Such simulations are typically run on resolutions of several kilometres and over decadal to millennial timescales.

5    Any melange parameterization needs to be combined with a large-scale calving parameterizations of which there are some. Benn et al. (2007) proposed a crevasse-depth calving-criterion assuming that once a surface crevasse reaches the water level, an iceberg calves off. This does not give a calving rate but rather the position of the calving front. It has been implemented in a flow-line model by Nick et al. (2010). Further calving parametrizations are a strain rate dependent calving rate for ice shelves (Levermann et al., 2012), a calving rate parametrization based on von Mises stress and glacier flow veloctiy (Morlighem et al., 10    2016) and a calving rate for a grounded glacier based on tensile failure (Mercenier et al., 2018).

In addition to calving caused by crevasses, another calving mechanism called cliff calving has first been proposed by Bassis and Walker (2011), who found that ice cliffs with a freeboard (ice thickness minus water depth) larger than $100\,\mathrm{m}$ are inherently unstable due to shear failure. Cliff calving was implemented as an almost step-like calving rate by Pollard et al. (2015); DeConto and Pollard (2016), while Bassis et al. (2017) implemented cliff calving as a criterion for the calving front position. 15    Finally, Schlemm and Levermann (2019) derived a cliff calving rate dependent on glacier freeboard and water depth by analyzing stresses close to the glacier terminus and using a Coulomb failure criterion.

Melange buttressing is likely to have a stabilizing effect on possible ice sheet instabilities. First, the so-called Marine Ice Sheet Instability (MISI) (Mercer, 1978; Schoof, 2007; Favier et al., 2014) can unfold if the grounding line is situated on a 20    reverse-sloping bed. Secondly, if the ice shelves buttressing the grounding line have disintegrated due to calving or melting and large ice cliffs become exposed, runaway cliff calving might lead to the Marine Ice Cliff Instability (MICI) (Pollard et al., 2015). DeConto and Pollard (2016) carried out past and future simulations of the Antarctic ice sheet with cliff calving implemented as a step function with a discussed but rather ad-hoc upper limit of $5\,\mathrm{km/a}$ as well as an additional hydrofracturing process that attacks the ice shelves. Edwards et al. (2019) did further analysis and compared the simulations of mid-Pliocene ice retreat (about 25    3 million years ago), where sea level was $5-20\,\mathrm{m}$ higher than present day, to observations. Given the uncertainty in many ice sheet parameters, uncertainties in air and ocean temperature forcing as well as uncertainty in determining Pliocene sea-level, agreement between simulations and observations could be achieved even without MICI. Calving rates larger than $5\,\mathrm{km/a}$ were not considered, but it is clear that using one of the recently derived calving parametrisations with calving rates up to at least $65\,\mathrm{km/a}$ (see fig. 1) would result in too much and too fast ice retreat. An upper limit on the calving rates appears to be necessary.

30

So far, the calving rate cutoff has been an ad-hoc assumption. However, this upper limit should correspond to some physical process that is responsible for limiting calving rates. We propose that ice mélange, a mix of icebergs and sea ice that is found in many glacial embayments, gives rise to a negative feedback on calving rates.

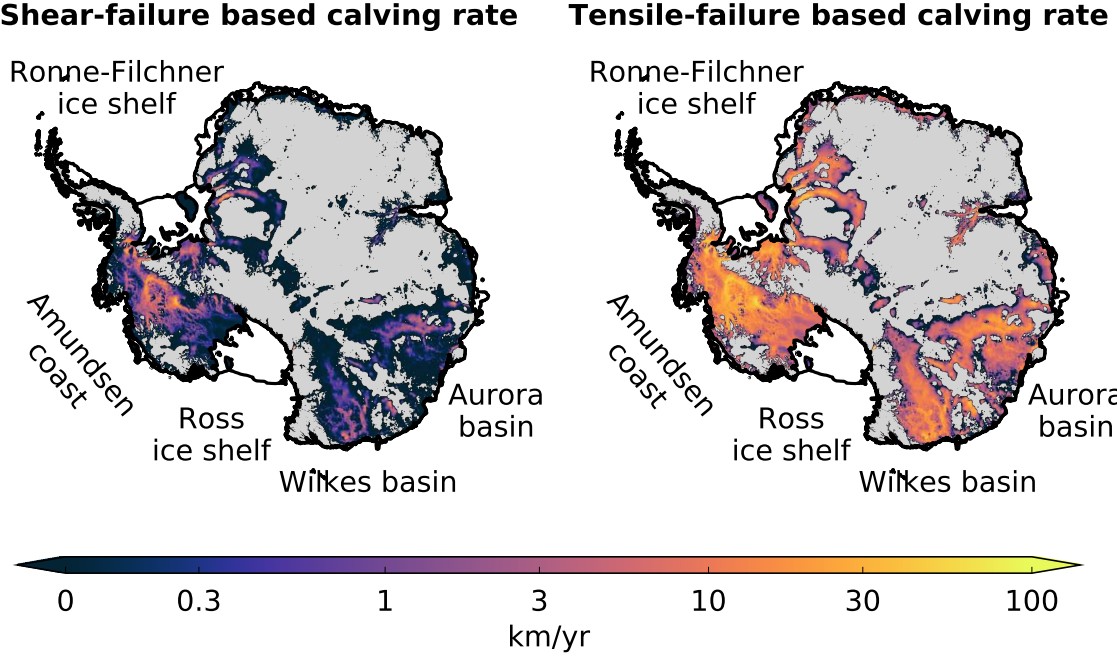

**Figure 1.** Potential shear-failure based calving rates (eq. 16) and tensile-failure based calving rates (eq. 15) in the grounded, marine regions of the Antarctic ice sheet. Floating ice is shown in white and grounded ice above sea level in grey. In the marine regions, ice is assumed to be at floatation thickness, which gives a minimal estimate of the potential calving rates. Estimates for shear calving rates go up to $65\,\mathrm{km/a}$ and estimates for tensile calving rates go up to $75\,\mathrm{km/a}$. If the grounding line retreat is faster than the speed with which the glacier terminus thins to floatation, calving rates could be even larger. Imposing an upper bound on the calving rates is necessary to prevent unrealistic, runaway ice loss.

Obeservations in Store glacier and Jakobshavn glacier in Greenland have shown that in the winter, when sea ice is thick, ice mélange prevents calving (Walter et al., 2012; Xie et al., 2019). This has also been reproduced in modelling studies of grounded marine glaciers (Krug et al., 2015; Todd et al., 2018, 2019): Backstresses from the mélange reduce the stresses in the glacier terminus thereby limiting crevasse propagation and reducing calving rates or preventing calving completely. There's a large uncertainty in the value of mélange backstresses, values given in the literature range between $0.02-3\,\mathrm{MPa}$ (Walter et al., 2012; Krug et al., 2015; Todd et al., 2018). Mélange backstress increases with $L/W$, the ratio of mélange length to the width of confining channel (Robel, 2017; Burton et al., 2018; Amundson and Burton, 2018). The presence of pinning points where the mélange grounds can also increase the backpressure. Seasonality of basal and surface melting and resulting thinning of the ice mélange is another important parameter for mélange backstress.

In addition to the reduced stresses caused by the backstress of the mélange, the presence of mélange may prevent a full-thickness ice berg from rotating away from the terminus, especially if the glacier is thicker than floatation thickness (Amundson et al., 2010). Tensile-failure based calving (Mercenier et al., 2018) is likely to produce full thickness icebergs and may

be hindered significantly by mélange. Shear-failure based calving (Schlemm and Levermann, 2019) is more likely to produce many smaller icebergs (breakup occurs through many small, interacting fractures at the foot of the terminus) and might be less influenced by mélange.

Ice mélange is also relevant for calving from ice shelves in Antarctica: the presence of mélange stabilizes rifts in the ice shelf
and can prevent tabular icebergs from separating from the iceshelf (Rignot and MacAyeal, 1998; Khazendar et al., 2009; Jeong et al., 2016).

We propose a negative feedback between calving rate and mélange thickness: A glacier terminus with high calving rates produces a lot of icebergs, which become part of the ice mélange in front of the glacier. The thicker the mélange is, the stronger
it buttresses the glacier terminus leading to reduced calving rates.

In section 2, we will show that with a few simple assumptions, this negative feedback between calving rate and mélange thickness leads to an upper limit on the calving rates. Section 3 shows that the model can extend beyond the steady-state. Application to two calving parametrizations and possible simplifications are discussed in section 4, and in section 5 the mélange buttressed calving rates are applied in an idealized glacier setup.

**2   Derivation of an upper limit to calving rates due to mélange buttressing**

Mélange can prevent calving in two ways: First, in the winter, additional sea ice stiffens and forcifies the mélange and can thus inhibit calving for example of Greenland glaciers (Amundson et al., 2010; Todd and Christoffersen, 2014; Krug et al., 2015). Secondly, a weaker mélange can still prevent a full-thickness iceberg from rotating out (Amundson et al., 2010) and thus prevent further calving.

Ice sheet models capable of simulating the whole Greenland or Antarctic ice sheet over decadal to millennial timescales cannot resolve the stresses at individual calving glacier termini and often do not resolve seasonal variations in forcing. Therefore, we need a model of mélange buttressed calving that is dependent on the geometries of the embayment and the ice sheet averaged over the year.

To this end, we start by assuming a linear relationship between mélange thickness and the reduction of the calving rate:

$$C = \left(1 - \frac{d_{cf}}{\gamma H}\right) C^*, \tag{1}$$

where $C^*$ is a calving rate derived for an unbuttressed glacier terminus (Morlighem et al., 2016; Mercenier et al., 2018; Schlemm and Levermann, 2019) and $C$ is the reduced calving rate caused by mélange buttressing. $H$ is the ice thickness at the glacier terminus and $d_{cf}$ is the mélange thickness at the calving front. In the absence of mélange, $d_{cf} = 0$, the calving
rate is not affected. As the mélange thickness increases, the calving rate is reduced, and when the mélange thickness equals a specific fraction $\gamma$ of the ice thickness $H$, calving is completely suppressed. The value of $\gamma$ may depend on the stiffness and compactness of the mélange and on how fractured the calving front is.

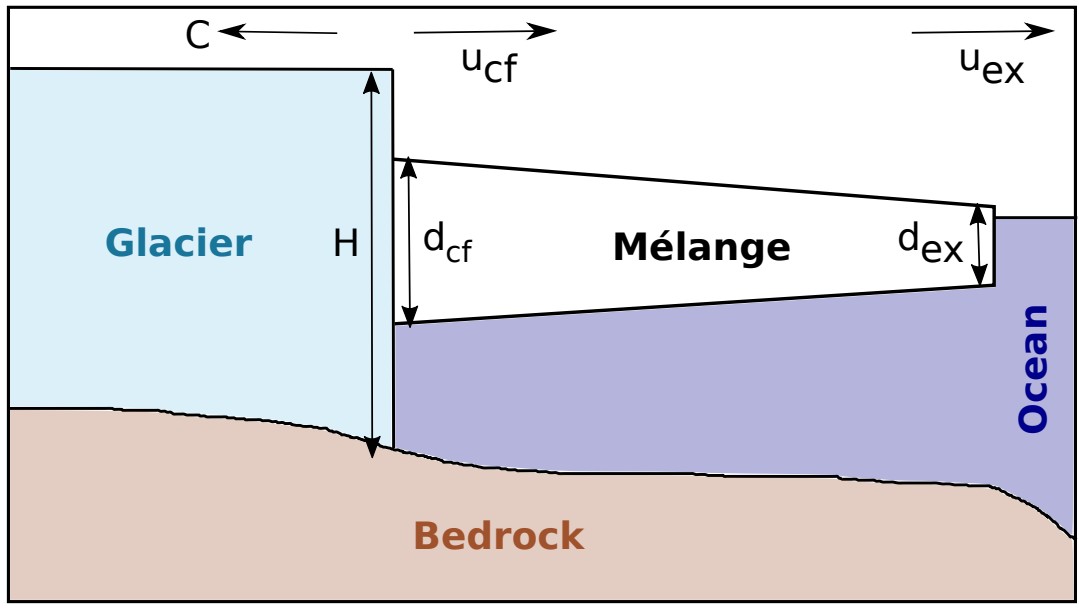

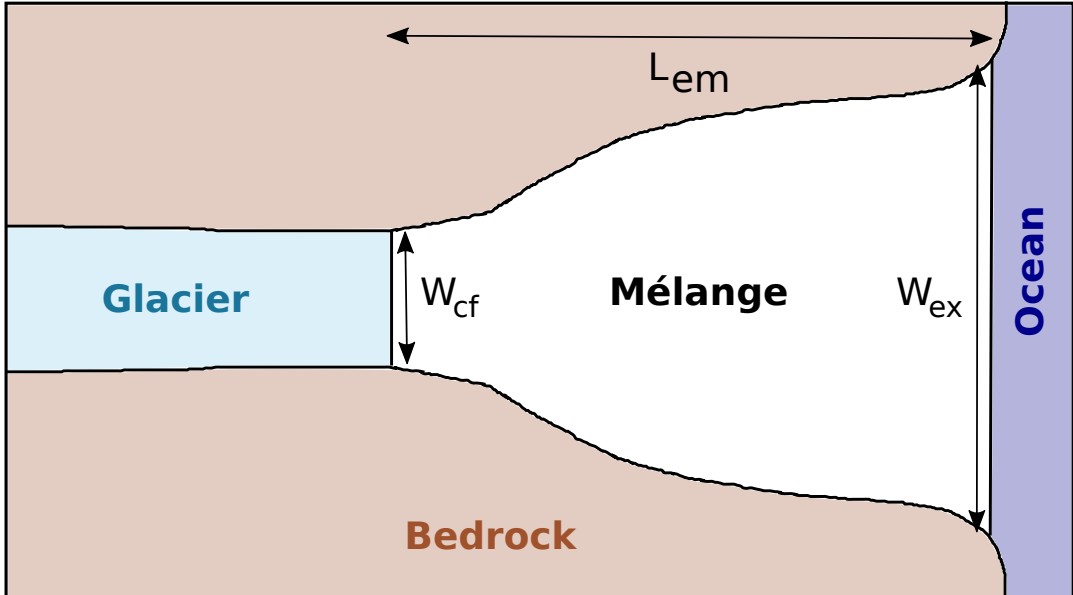

**Figure 2.** Geometry of the glacier terminus, ice mélange and embayment as a side view and a top view. The side view shows the ice thickness $H$, the calving front thickness $d_{cf}$ and exit thickness $d_{ex}$ of the ice mélange as well as the calving rate $C$ and the mélange exit velocity $u_{ex}$. The plan view shows the embayment width at the calving front $W_{cf}$ and the embayment exit width $W_{ex}$ as well as the length of the embayment $L_{em}$. The mélange does not necessarily need to extend all the way to the embayment exit: if it is shorter, then $L_{em}$ donates the mélange length and $W_{ex}$ the width of the embayment at the position, where the mélange ends.

| | |
|---:|:---|
| $H$ | ice thickness |
| $C^*, C$ | unbuttressed and buttressed calving rates |
| $\gamma$ | fraction of the ice thickness |
| $d_{cf}$ | mélange thickness at the calving front |
| $d_{ex}$ | mélange thickness at the embayment exit |
| $\bar{d}$ | average mélange thickness |
| $V$ | mélange volume |
| $W_{cf}$ | embayment width at the calving front |
| $W_{ex}$ | embayment width at the embayment exit |
| $\bar{W}$ | average embayment width |
| $L_{em}$ | embayment (mélange) length |
| $A_{em}$ | embayment (mélange) area |
| $u_{cf}$ | ice flow velocity at the calving front |
| $u_{ex}$ | mélange exit velocity |
| $m$ | average mélange melt rate |
| $\beta$ | mélange thinning gradient |
| $\mu_0$ | mélange internal friction |
| $d_m$ | mélange thickness lost due to melting |
| $a$ | inverse of $C_{max}$ |
| $C_{max}$ | upper limit on calving rates |

**Table 1.** Overview over the variables used in sec. 2. The embayment and mélange geometry is illustrated in fig. 2

In order to estimate the mélange thickness at the calving front, $d_{cf}$, we assume a glacier terminating in an embayment already filled with ice mélange, where the mélange does not necessarily need to extend all the way to the embayment exit. Furthermore, we assume that the mélange properties are constant over the entire embayment and that the mélange thickness thins linearly along the flow direction (fig. 2).

5 The embayment area is given by $A_{em}$, its width at the calving front by $W_{cf}$ and its width at the exit by $W_{ex}$. The calving rate $C$ is assumed to be equal to the ice flow $u_{cf}$ so that the calving front remains at a fixed position. As the mélange thins on its way to the embayment exit, it has an exit thickness $d_{ex}$ and an exit velocity $u_{ex}$ at which mélange and icebergs are transported away by ocean currents. (See also table 1.)

We consider a mélange volume $V = A_{em}\bar{d}$, where $\bar{d}$ is the average mélange thickness. The overall rate of change of the mélange

10 volume is given by:

$$\frac{\mathrm{d}V}{\mathrm{d}t} = W_{cf}HC - W_{ex}d_{ex}u_{ex} - mA_{em} \tag{2}$$

where the first term corresponds to mélange production at the calving front, the second term corresponds to mélange exiting into the ocean and the third term corresponds to mélange loss through melting (assuming an average melt rate $m$). Assuming a

steady state of mélange production and loss resulting in a constant mélange geometry ($\mathrm{d}V/\mathrm{d}t = 0$), we can solve eq. 2 for $d_{ex}$:

$$d_{ex} = \frac{W_{cf}HC - mA_{em}}{W_{ex}u_{ex}} \tag{3}$$

This equation only has a physical solution if $mA_{em} < W_{cf}HC$, which implies that melting is small enough that mélange

actually reaches the embayment exit. If this is not given, mélange may still exist but it will not reach the embayment exit and the above inequality becomes a condition on the mélange length. Assuming a viscoplastic rheology and quasi-static flow of ice mélange, Amundson and Burton (2018) found that mélange thinning along the embayment length is given by an implicit exponential function. A linear approximation gives

$$d_{cf} = \beta d_{ex}, \qquad \beta = b_0 + b_1\mu_0 L_{em}/\bar{W} \tag{4}$$

where $\mu_0$ is the internal friction of the mélange, $b_0$ and $b_1$ are constants slightly larger than 1 and $\bar{W}$ is the average embayment width (for more details see appendix A). Then the mélange thickness at the calving front is given as

$$d_{cf} = aCH - d_m, \qquad \text{with} \quad a = \frac{W_{cf}}{W_{ex}}\frac{\beta}{u_{ex}}, \quad d_m = \beta\frac{mA_{em}}{W_{ex}u_{ex}} \tag{5}$$

$d_m$ is the mélange thickness lost to melting, $a$ has the units of an inverse calving rate and will be related to the upper bound on calving rates in eq. 7. Inserting eq. 5 into eq. 1, we get

$$C = \left(1 + \frac{d_m}{\gamma H}\right)\frac{C^*}{1 + \tilde{a}C^*}, \qquad \text{with} \quad \tilde{a} = a\gamma^{-1} \tag{6}$$

Neglecting melting for simplicity we get

$$C = \frac{C^*}{1 + \tilde{a}C^*} = \frac{C^*}{1 + C^*/C_{max}} \tag{7}$$

This function is linear, $C \approx C^*$, for small unbuttressed calving rates ($C^* \ll C_{max} = \tilde{a}^{-1}$) and the buttressed calving rate $C$ saturates at an upper limit $C_{max} = \tilde{a}^{-1}$ for large unbuttressed calving rates ($C^* \gg C_{max} = \tilde{a}^{-1}$). This means that the param-

eter $\tilde{a}$ can be considered as the inverse maximum calving rate, $C_{max} = \tilde{a}^{-1}$, which is dependent on the embayment geometry, mélange flow properties and the embayment exit velocity. If the unbuttressed calving rate, $C^*$, is small compared to the upper bound $C_{max}$, there is little buttressing. If $C^*$ is of the same order of magnitude or larger than $C_{max}$, there is significant buttressing (see fig. 5). Including melt of the mélange leads to higher calving rates, because melting thins the mélange and weakens the buttressing it provides to the calving front.

    Rather than imposing an upper bound on the calving rates as an ad hoc cut-off as done by DeConto and Pollard (2016); Edwards et al. (2019), mélange buttressing gives a natural upper bound on the calving rate which is reached smoothly. The value of the upper bound can be different for each glacier, depending on the embayment geometry, and may change seasonally in accord with mélange properties.


According to eq. 5 and eq. 7, the upper limit on calving rates is a function of embayment geometry and mélange properties,

$$C_{max} = \frac{W_{ex}}{W_{cf}} \left( b_0 + b_1 \mu_0 \frac{L_{em}}{\bar{W}} \right)^{-1} \gamma \, u_{ex} \tag{8}$$

Since $C_{max}$ is proportional to $W_{ex}/W_{cf}$, embayments that become narrower at some distance from the calving front experience stronger mélange buttressing and consequently have smaller upper limits than embayments that are widening towards the ocean. Also the longer the embayment is compared to the average embayment width ($L_{em}/\bar{W}$), the smaller the upper limit is, even though friction between the mélange and the embayment walls has not been taken explicitly into account. Previous studies have already shown this for the mélange backstress (Burton et al., 2018; Amundson and Burton, 2018). Fast ocean currents or strong wind forcing at the embayment exit may lead to fast export of mélange (fast exiting velocities $u_{ex}$) and hence reduced mélange buttressing. Melting of the mélange from below will also reduce mélange buttressing and hence increase $C_{max}$. The stronger the internal friction of the mélange ($\mu_0$), the larger the buttressing effect.

It can be instructive to consider the force per unit width at the calving front as given by eq. (10) in Amundson and Burton (2018) with the mélange thickness given by eq. 5 derived above:

$$\frac{F}{W} = \frac{1}{2} \rho_i \left( 1 - \frac{\rho_i}{\rho_w} \right) \left( 1 - \frac{d_{ex}}{d_{cf}} \right) d_{cf}^2 \tag{9}$$

$$= \frac{1}{2} \rho_i \left( 1 - \frac{\rho_i}{\rho_w} \right) \left( b_0 + b_1 \mu_0 \frac{L_{em}}{\bar{W}} - 1 \right) \left( \frac{HC^*}{1 + C^* \frac{W_{cf}}{W_{ex} \gamma u_{ex}} \left( b_0 + b_1 \mu_0 \frac{L_{em}}{\bar{W}} \right)} \frac{W_{cf}}{W_{ex} u_{ex}} - \frac{m A_{em}}{W_{ex} u_{ex}} \right)^2$$

## 3   Beyond a steady-state solution

The mélange buttressing model derived in section 2 assumes mélange to be in a steady state with a fixed mélange geometry. This implies a fixed calving front position. This assumption is not fulfilled if glacier retreat is considered. Therefore it is worthwhile to go beyond the steady-state solution.

If the mélange geometry changes in time, the change in the mélange volume can be expressed as:

$$\frac{dV}{dt} = \frac{d}{dt} \int_0^{L(t)} dx \, W(x) \, d(x,t) \tag{10}$$

where $L(t)$ is the distance between the the embayment exit and the calving front, $W(x)$ the width of the embayment at a distance $x$ from the embayment exit, $d(x,t)$ is the mélange thickness and the embayment exit is fixed at $x = 0$. This expression is equal to the sum of mélange production and loss terms given in eq. 2. By applying the Leibniz integral rule to the volume integral of eq. 10 as well as rewriting the mélange production and loss terms as functions of time and calving front position, eq. 2 becomes

$$W_L H C - W_0 d_0 u_{ex} - m \int_0^L dx \, W(x) = W_L \beta d_0 \cdot \frac{d}{dt} L + \left( \int_0^L dx \, W(x) \right) \cdot \frac{d}{dt} (\beta d_0) \tag{11}$$

with $L = L(t)$, $H = H(L(t))$, $C = C(t)$, $d_0 = d(0,t)$, $W_0 = W(0)$, $W_L = W(L(t))$ and $\beta = \beta(L(t))$. The first three terms on the left hand side are the mélange production through calving, the mélange loss at the embayment exit and the mélange melting, respectively, and the right hand side is the rewritten volume integral. If the embayment geometry $W(x)$ as well as the ice thickness at the calving front $H(L(t))$ are known, the calving rate $C(t)$ is given by

$$C(t) = \left(1 - \frac{\beta(L(t))d(0,t)}{\gamma H(L(t))}\right) C^* \tag{12}$$

and an equation for the evolution of the mélange length, $L(t)$ is assumed, this differential equation for $d(0,t)$ can be solved. We will consider two cases for the evolution of $L(t)$: first, a constant mélange length where the mélange retreats with the calving front, and second, mélange pinned to the embayment exit so that the mélange length grows with the rate of the glacier retreat. We now consider an idealized setup with constant ice thickness, $H(x) = H$, as well as constant embayment width, $W(x) = W$. Eqs. 11 - 14 are solved numerically for the parameter values $H = 1000\,\mathrm{m}$, $W = 10\,\mathrm{km}$, $\mu = 0.3$, $\gamma = 0.2$, $C^* = 3\,\mathrm{km/a}$, $u_{ex} = 100\,\mathrm{km/a}$, $b_0 = 1.11$, $b_1 = 1.21$, and the initial conditions $L(0) = 10\,\mathrm{km}$ and $d(0) = 10\,\mathrm{m}$. We consider a scenario without mélange melting, $m = 0$, and a scenario with mélange melting, where the melt rate is set to $m = 10\,\mathrm{m/a}$.

## 3.1 Constant mélange length

First, we will assume a constant mélange length:

$$\frac{\mathrm{d}}{\mathrm{d}t}L(t) = 0 \tag{13}$$

This might be either because the calving front does not move (ice flow equals calving rate) or because the mélange is not pinned to the embayment exit and retreats with the calving front, keeping a constant length.

The solutions for the force per unit width at the calving front, $F(t)/W$, mélange thickness at the embayment exit, $d(0,t)$, mélange thickness at the calving front, $d(L(t),t)$, and the resulting buttressed calving rate, $C(t)$ are shown in fig. 3. The initial conditions chosen do not correspond to a steady-state solution, but the mélange equilibrates quickly, with the free evolution solution reaching the constant steady state solution in less than six months of simulation time. If melting is included, the mélange is thinner and hence the final calving rate is slightly larger.

The force per unit width is small compared to other mélange models (Amundson and Burton, 2018; Burton et al., 2018), but it's not an integral part of the model, rather only a diagnostic. A force of about about $10^7\,\mathrm{N\,m^{-1}}$ (Amundson et al., 2010) prevents icebergs from rotating out and would inhibit calving. A weaker mélange merely reduces calving rates as seen here. Also the setup here is of a rather short mélange ($L/W = 1$) and hence the mélange is not very thick.

## 3.2 Mélange pinned to embayment exit

Second, we will assume that the mélange is pinned to the embayment exit, hence the mélange length grows with the rate of glacier retreat:

$$\frac{\mathrm{d}}{\mathrm{d}t}L(t) = C(t) - u_{cf}(t) \tag{14}$$

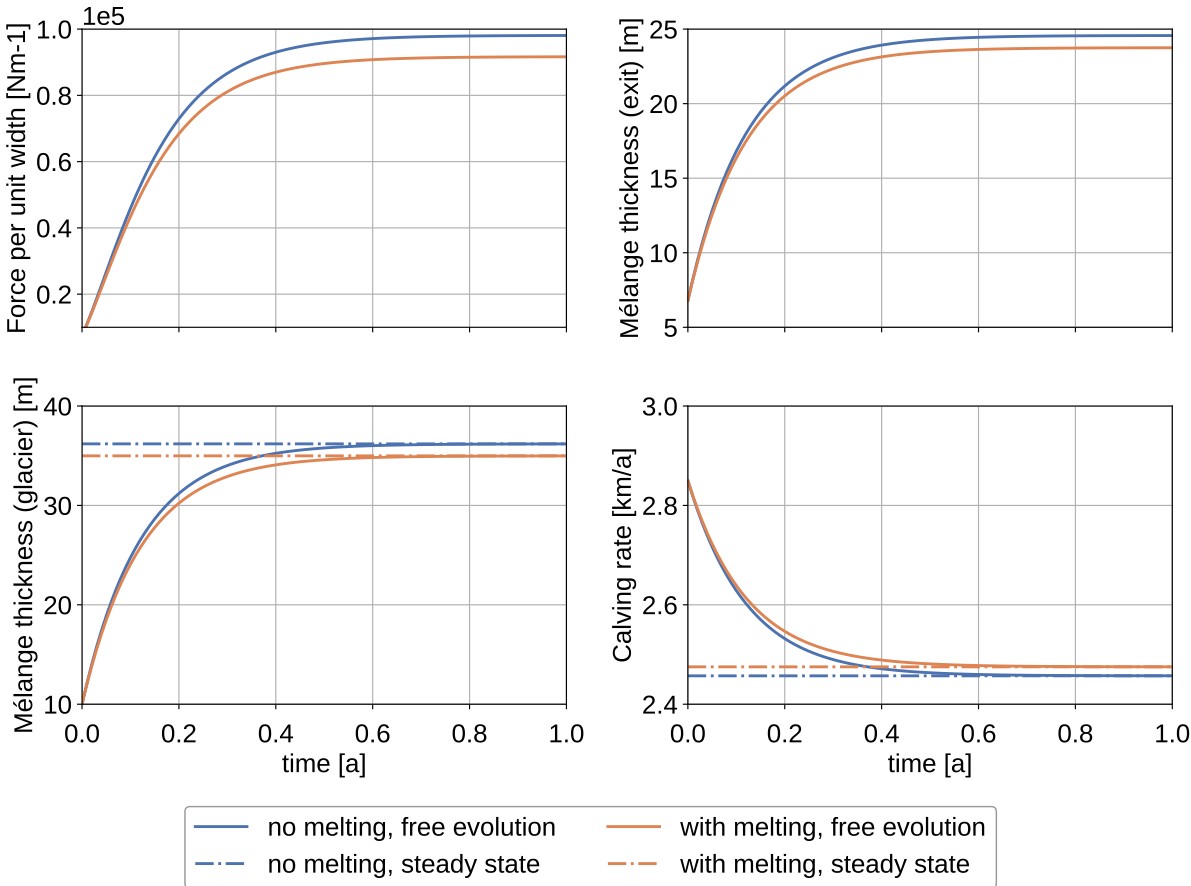

**Figure 3.** Top row panels show the numerical solutions of force per unit width, $F(t)/W$, and mélange thickness at the embayment exit, $d(0,t)$, given by eqs. 11 - 14 if mélange length is assumed to be constant. Two scenarios are considered: without melting (blue line) and with melting (orange). The bottom panels show the mélange thickness at the calving front, $d(L(t),t)$, and the resulting buttressed calving rate, $C(t)$. The solution with free evolution of the mélange geometry (continuous line) is contrasted with the steady-state solution obtained by plugging the mélange length, $L(t)$, into eq. 5 and 6, respectively, (dashed line), showing equilibration of the mélange in less than a year.

where the ice flow velocity at the calving front, $u_{cf}(t)$, depends on the bed topography and the ice dynamics. In this simplified setup, we will neglect ice flow by setting $u_{cf} = 0$

The solutions for mélange length, $L(t)$, mélange thickness at the embayment exit, $d(0,t)$, mélange thickness at the calving front, $d(L(t),t)$, and the resulting buttressed calving rate, $C(t)$ are shown in fig. 4. In the scenario without melting, mélange 5 length and thickness at the calving front increase, while mélange thickness at the embayment exit and buttressed calving rate decrease. If melting of mélange is considered, the mélange thickness at the calving front increases initially, and then decreases until the embayment is mélange-free, since the volume of mélange melted increases with mélange area.

A comparison between these solutions, where the mélange geometry is free to evolve, and the corresponding steady-state

solution for mélange thickness at the calving front and the calving front, obtained by plugging the mélange length, $L(t)$, into eq. 5 and 6, respectively, shows good agreement (see bottom panels of fig. 4). As in the previous example the mélange equilibrates quickly, and the free evolution solution follows the steady state solution closely in the remaining time. This justifies the adaptive approach discussed in section 5.2.

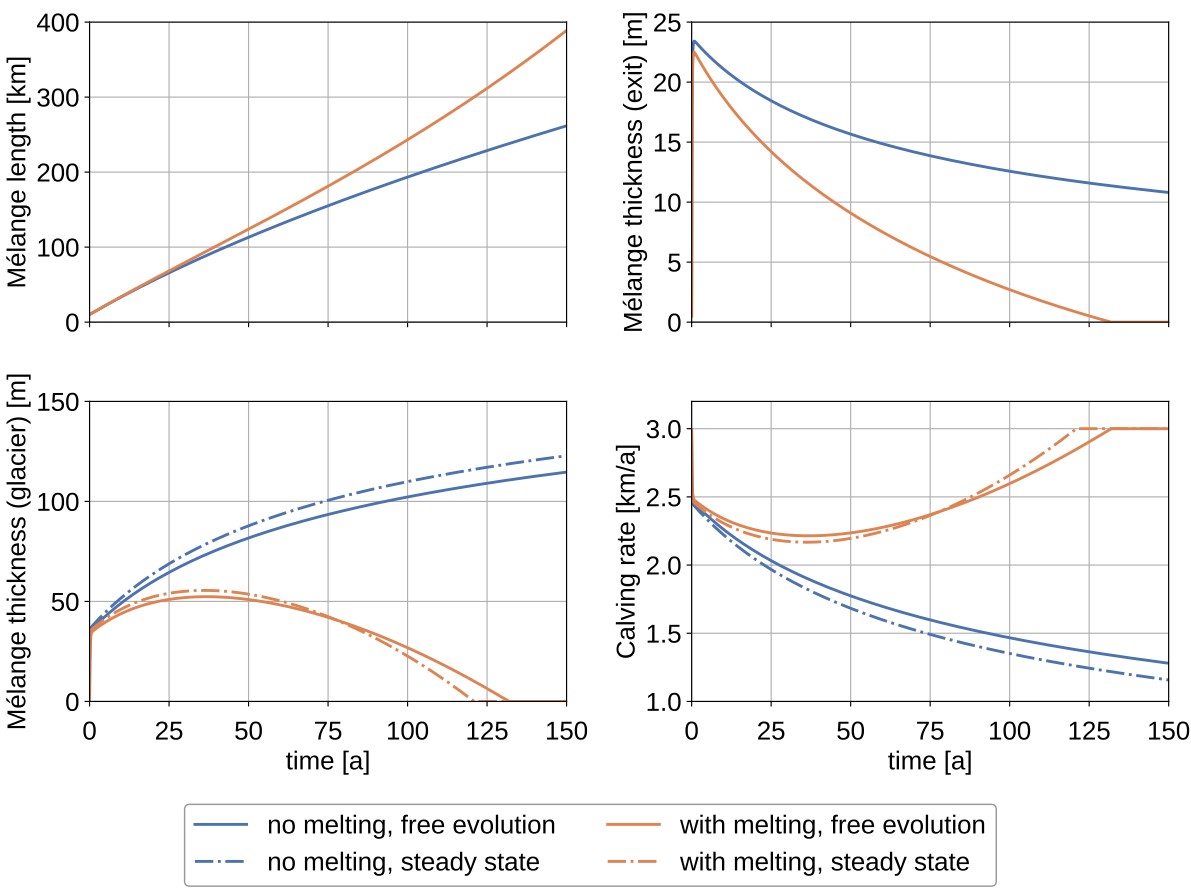

**Figure 4.** Top row panels show the numerical solutions of mélange length, $L(t)$, and mélange thickness at the embayment exit, $d(0,t)$, given by eqs. 11 - 14.. Two scenarios are considered: without melting (blue line) and with melting (orange). The bottom panels show the mélange thickness at the calving front, $d(L(t),t)$, and the resulting buttressed calving rate, $C(t)$. The solution with free evolution of the mélange geometry (continuous line) is contrasted with the steady-state solution obtained by plugging the mélange length, $L(t)$, into eq. 5 and 6, respectively, (dashed line).

## 4 Application to stress-based calving parametrizations

Bassis and Walker (2011) showed that ice cliffs with a glacier freeboards (ice thickness minus water depth) exceeding $\approx 100\,\mathrm{m}$ are inherently unstable due to shear failure. However, smaller ice cliffs calve off icebergs as well. Mercenier et al. (2018) derived a tensile-failure based calving parametrization for calving fronts with freeboards below this stability limit, while Schlemm and Levermann (2019) derived a shear-failure based calving parametrization for calving fronts with freeboards exceeding the stability limit.

### 4.1 Tensile-failure based calving

A calving relation based on tensile failure was derived by Mercenier et al. (2018) who used the Hayhurst stress as failure criterion to determine the position of a large crevasse that would separate an iceberg from the glacier terminus and calculated the timescale of failure using damage propagation. The resulting tensile calving rate is given by

$$C_t^* = B \cdot \left(1 - w^{2.8}\right) \cdot \left((0.4 - 0.45(w - 0.065)^2) \cdot \rho_i gH - \sigma_{th}\right)^r \cdot H \tag{15}$$

with effective damage rate $B = 65\,\mathrm{MPa}^{-r}\mathrm{a}^{-1}$, stress threshold for damage creation $\sigma_{th} = 0.17\,\mathrm{MPa}$, constant exponent $r = 0.43$, ice density $\rho_i = 1020\,\mathrm{kg m}^{-3}$, gravitational constant $g = 9.81\,\mathrm{ms}^{-2}$ and the relative water depth, $w = D/H$. This calving relation was derived for glacier fronts with a glacier freeboard smaller than the stability limit.

### 4.2 Shear-failure based calving

An alternative calving relation based on shear failure of an ice cliff was derived in Schlemm and Levermann (2019), where shear failure was assumed in the lower part of an ice cliff with a freeboard larger than the stability limit. The resulting shear calving rate is given by:

$$C_s^* = C_0 \cdot \left(\frac{F - F_c}{F_s}\right)^s \tag{16}$$

$$F_s = \left(114.3\,(w - 0.3556)^4 + 20.94\right)\,\mathrm{m} \tag{17}$$

$$F_c = (75.58 - 49.18w)\,\mathrm{m} \tag{18}$$

$$s = 0.1722 \cdot \exp(2.210w) + 1.757 \tag{19}$$

with relative water depth $w \equiv D/H < 0.9$ and glacier freeboard $F \equiv H - D = H \cdot (1 - w)$. $F_c$ is the critical freeboard above which calving occurs, $F_s$ is a scaling parameter and $s$ a nonlinear exponent. The scaling parameter $C_0$ is given as $C_0 = 90\,\mathrm{m}\,\mathrm{a}^{-1}$, but this value is badly constrained and therefore $C_0$ can be considered a free parameter which parametrizes the uncertainty in the time to failure. This calving law assumes that there is no calving for freeboards smaller than the critical freeboard $F < F_c$.

Plugging the calving relation, eq. 16, into the mélange buttressed calving rate given by eq. 7 and expanding, it can be shown that the value of the upper bound $C_{max}$ has a greater influence on the resulting calving rates than the scaling parameter $C_0$: Let's call the dimensionless freeboard-dependent part of the cliff calving relation

$$\tilde{C}_s = \left( \frac{F - F_c}{F_s} \right)^s , \tag{20}$$

then the buttressed calving rate is

$$C_s = \frac{\tilde{C}_s}{\frac{1}{C_0} + \frac{\tilde{C}}{C_{max}}} \tag{21}$$

Then if $1 \ll \tilde{C}$

$$C_s = C_{max} - \frac{C_{max}^2}{\tilde{C} C_0} \tag{22}$$

For small $\tilde{C}$ the choice of scaling parameter $C_0$ influences the final calving rate $C$, but for large $\tilde{C}$, the upper bound $C_{max}$ determines the resulting calving rate. Since the scaling parameter $C_0$ is difficult to constrain and has little influence on the mélange buttressed calving rate, it makes sense to use a fixed value, e.g. $C_0 = 90\,\mathrm{m\,a}^{-1}$, and treat only the upper bound $C_{max}$ as a free parameter (which is dependent on the embayment geometry and mélange properties).

### 4.3 Comparison of the calving parametrizations

A comparison of the two stress-based calving rates can be divided into four parts (see fig. 5a):

1. According to the calving parametrisations considered here (eq. 15 and eq. 16), glacier fronts with very small freeboards ($< \approx 20\,\mathrm{m}$) do not calve.

2. For glacier freeboards below the stability limit of $\approx 100\,\mathrm{m}$, there is only tensile calving with calving rates up to $\approx 10\,\mathrm{km/a}$ and no shear calving.

3. Above the stability limit, shear calving rates increase slowly at first but speed up exponentially and equal the tensile calving rates at freeboards between $200 - 300\,\mathrm{m}$ and calving rates between $15 - 60\,\mathrm{km/a}$. There is a spread in these values because both calving rates depend on the water depth as well as the freeboard.

4. For even larger freeboards, shear calving rates have a larger spread than tensile calving rates and much larger values for cliffs at floatation.

A comparison of the buttressed calving rates can be classified in the same way (see fig. 5b-d) where the only difference is that large calving rates converge to a value just below the upper limit $C_{max}$ and hence the difference between tensile and shear calving rates for large freeboards is smaller.

Summarizing, there are two different calving parametrizations, based on tensile and shear failure and derived for glacier freeboards below and above the stability limit, respectively. It might seem obvious that one should simply use each calving law in the range for which it was derived. However, that would lead to a large discontinuity in the resulting calving rate because the tensile calving rate is much larger at the stability limit than the shear calving rate. Another possibility is to use each parametrization in the range for which it gives the larger calving rate. Since it is likely that in nature large ice cliffs fail due to a combination of failure modes, it also seems reasonable to use a combination of tensile and shear calving rates.

In the context of the Marine Ice Cliff Instability (MICI) hypothesis, one would expect a sudden and large increase in calving rates for ice cliffs higher than the stability limit. Despite a nonlinear increase of calving rates in the unbuttressed case, neither of the two stress-based calving parametrizations (Mercenier et al., 2018; Schlemm and Levermann, 2019) nor a combination of them shows discontinuous behaviour at the stability limit.

## 4.4 Simplified calving relations

There are uncertainties in both calving laws because a dominating failure mode is assumed (shear and tensile failure, respectively), while in reality failure modes are likely to interact. Also, in the calving laws ice is assumed to be previously undamaged, whereas a glacier is usually heavily crevassed and therefore weakened near the terminus. In addition, shear calving has a large uncertainty with respect to the time to failure which leads to uncertainty in the scaling parameter $C_0$. These uncertainties together with the observation that the upper limit $C_{max}$ seems to have a stronger influence on resulting calving rates than the choice of calving law provide a good reason to consider simplifying these calving laws.

The important distinction between shear and tensile calving is that shear calving has a much larger critical freeboard: for small freeboards ($F < 100\,\mathrm{m}$), we have tensile calving but no shear calving. Since the mélange buttressed calving rate is linear in the calving rates for small calving rates, this distinction remains in the buttressed calving rates (see fig. 5). However, for larger freeboards the calving rates approach the upper limit no matter which calving law was chosen. This distinction should be conserved in the simplified calving relations.

The dependence of the calving rate on water depth is important in the unbuttressed case (see fig. 5a): there's a large range between calving rates for the same freeboard and different relative water depths – that's because larger relative water depth implies a larger overall depth. For the same glacier freeboard, this means a larger ice thickness and therefore larger stresses in the ice column, implying a larger calving rate. But in the mélange buttressed case, large calving rates are more strongly buttressed than small calving rates. Thus the large range of possible calving rates for a given glacier freeboard is transformed into a much smaller range, so that water depth becomes less important. (see fig. 5b-d)

Therefore we consider simplifications of the calving relations where we average over the water depth and further simplify. This is done mostly for illustrative purposes.

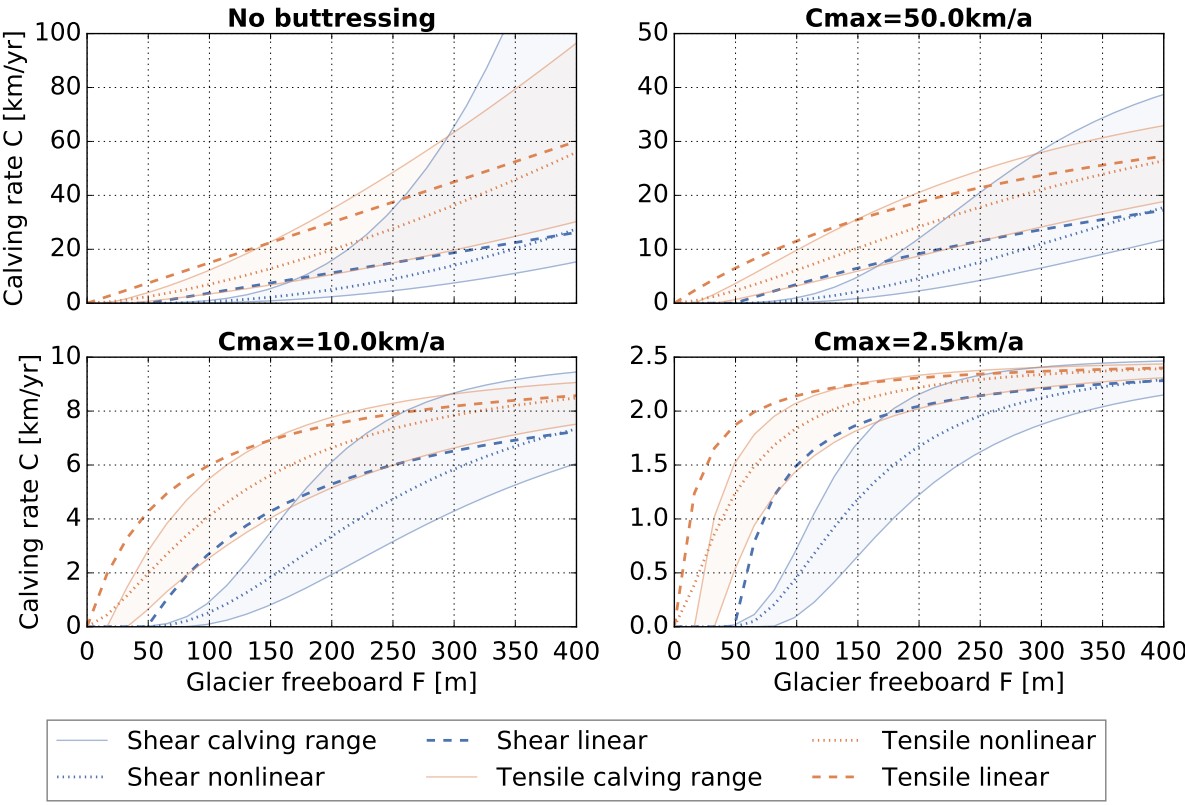

**Figure 5.** Calving rates as a function of glacier freeboard (ice thickness - water depth) in the unbuttressed case and for a range of upper bounds $C_{max}$. Shear calving and tensile calving rates depend also on the water depth: Two lines are shown for each configuration, the lower line for a dry cliff ($w = 0.0$) and the upper line for a cliff at floatation ($w = 0.8$). This spans the range of possiible calving rates for a given freeboard. Also shown are the nonlinear (dotted line) and linear (dashed lines) approximations to these calving laws. In the tensile case, calving commences with freeboard F=0, while shear calving only happens for freeboards larger $F_c \approx 50\,\text{m}$.

Take the shear calving relation:

$$C_s^* = C_0 \cdot \left( \frac{F - F_c}{F_s} \right)^s \tag{23}$$

where $C_0 = 90\,\text{m/a}$, $s(w) \in [1.93, 3.00]$, $F_c(w) \in [30.9, 75.0]\,\text{m}$ and $F_s(w) \in [21.0, 31.1]\,\text{m}$. In choosing round values within these intervals, we can simplify the relation.

$$5 \quad C_{s,nonlin}^* = 90\,\text{m/a} \cdot \left( \frac{F - 50\,\text{m}}{20\,\text{m}} \right)^2 \tag{24}$$

Because the exponent $s$ is on the smaller end of the possible values we chose a smaller value for $F_s$ to get an approximation that lies well within the range of the full cliff calving relation, though it lies at the lower end (see fig. 5). An even simpler linear

approximation

$$C^*_{s,lin} = 75\,\mathrm{a}^{-1} \cdot (F - 50\,m) \tag{25}$$

overestimates the calving rates for small freeboards ($F < 200\,\mathrm{m}$) and underestimates for large freeboards ($F > 600\,\mathrm{m}$).

The tensile calving relation can be written as

$$C^*_t = a(w)\,(b(w)F - \sigma_{th})^{0.43} \cdot F \approx c \cdot F^{1.5} \tag{26}$$

and can be fitted with a power function

$$C^*_{t,nonlin} = 7\,\mathrm{m}^{-0.5}\mathrm{a}^{-1} \cdot F^{1.5} \tag{27}$$

or a linear function

$$C^*_{t,lin} = 150\,\mathrm{a}^{-1} \cdot F \tag{28}$$

Here we neglect the small offset in freeboard that tensile calving has. This gives us two kinds of simplified calving relations to compare: one that begins calving immediately and one that only calves off cliffs larger than a certain critical freeboard. For both we have a linear approximation that overestimates small calving rates, and a nonlinear approximation that lies well within the original spread of calving rates (see fig. 5).

## 5   Mélange buttressed calving in an idealized glacier setup

We consider a MISMIP+-like glacier setup (Cornford et al., 2020), that is symmetric about $x = 0$ and has periodic boundary conditions on the fjord walls. The glacial valley has an average bedrock depth of $200\,\mathrm{m}$ and a width of $40\,\mathrm{km}$ and experiences a constant accumulation of $1.5\,\mathrm{m/a}$ (see fig. 6). The setup has rocky fjord walls and where the bedrock wall is below sea level, there is grounded ice resting on it. This grounded ice does not retreat during the calving experiments and forms the embayment.

Ice flow is concentrated in the middle of the channel where the bedrock is significantly deeper. Since there is no ice reservoir at the top of the glacier, this setup can also be considered as a model for a mountain glacier.

The experiments were done with the Parallel Ice Sheet Model (PISM) (Bueler and Brown, 2009; Winkelmann et al., 2011) which uses the shallow ice approximation (Hutter, 1983) and the shallow shelf approximation (Weis et al., 1999). We use Glen's flow law in the isothermal case and a pseudoplastic basal friction law (the PISM authors, 2018).

A spin-up simulation was run until it reached a steady state configuration with an attached ice shelf. During the experiment phase of the simulation all floating ice is removed at each timestep. When the ice shelf is removed, the marine ice sheet instability (MISI) kicks in because of the slightly retrograde bed topography and the glacier retreats. Calving accelerates this retreat. Experiments were made with no calving (MISI only), mélange buttressed shear calving and its nonlinear and linear approximation as well as mélange buttressed tensile calving and its two approximations. The inital upper bound was varied

$C_{max} = [2.5, 10.0, 50.0, 500.0]\,\mathrm{km/a}$ where the last upper bound was chosen to be large enough that the calving rates nearly match the unbuttressed calving rates.

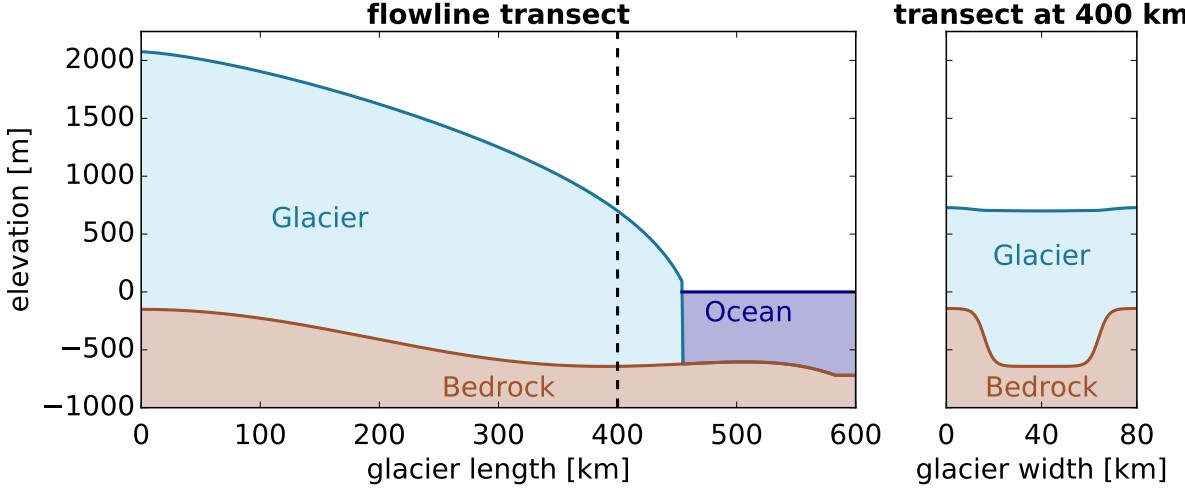

**Figure 6.** Setup of the idealized glacier experiments. Only one half of the setup is shown, the glacier is connected to an identical copy at the left to ensure periodic boundary conditions at the ice divide.

### 5.1 Constant upper bound on calving rates

In this experiment, the upper bound was kept constant even though the glacier retreated and embayment length increased. The buttressing eq. 7 was derived assuming a steady-state mélange geometry which implies a fixed mélange geometry. This is the case in this idealised setup, if we assume that mélange length is fixed and mélange retreats with the calving front, as in sec. 3.1.

Fig. 7 shows the simulated glacier retreat. Even without calving in the MISI only experiment, there is a significant retreat after removing the ice shelves because of the buttressing loss and slightly retrograde bed of the glacier. The glacier retreats from a front position at 440km to 200km in the first 100 years, after which the retreat decelerates and the glacier stabilizes at a length of about 130km. Adding calving leads to additional retreat: the higher the upper bound on the calving rates, the faster the retreat.

Shear calving causes less additional retreat than tensile calving because it has small calving rates for freeboards below 150m. Since the channel is rather shallow the freeboards are generally small. Only the linear approximation of shear calving has a significant ice retreat because even though it starts only with a freeboard of 50m, it grows much faster than the actual shear calving or the nonlinear approximation. But it also reaches a stable glacier position when the ice thickness is smaller than the critical freeboard condition.

The assumption of tensile calving causes the glacier to retreat much faster. The linear approximation, which has higher calving rates for small freeboards, leads to a faster retreat. For the nonlinear approximation the glacier is close to floatation for most of its retreat which corresponds to the upper half of the tensile calving range. This approximation gives smaller calving rates and hence slower retreat. None of the tensile calving relations allow the glacier to stabilize. That is to say the minimum freeboard

below which an ice front is stable for shear calving is ultimately the stabilizing factor in these simulations.

Fig. 8 shows that the effect of mélange buttressing becomes relevant for small values of the export of ice out of the embayment, i.e. for small values of $C_{max}$. In this limit of strong buttressing, i.e. where the parameterization of equation 7 is relevant, the glacier retreat becomes almost independent of the specific calving parameterization.

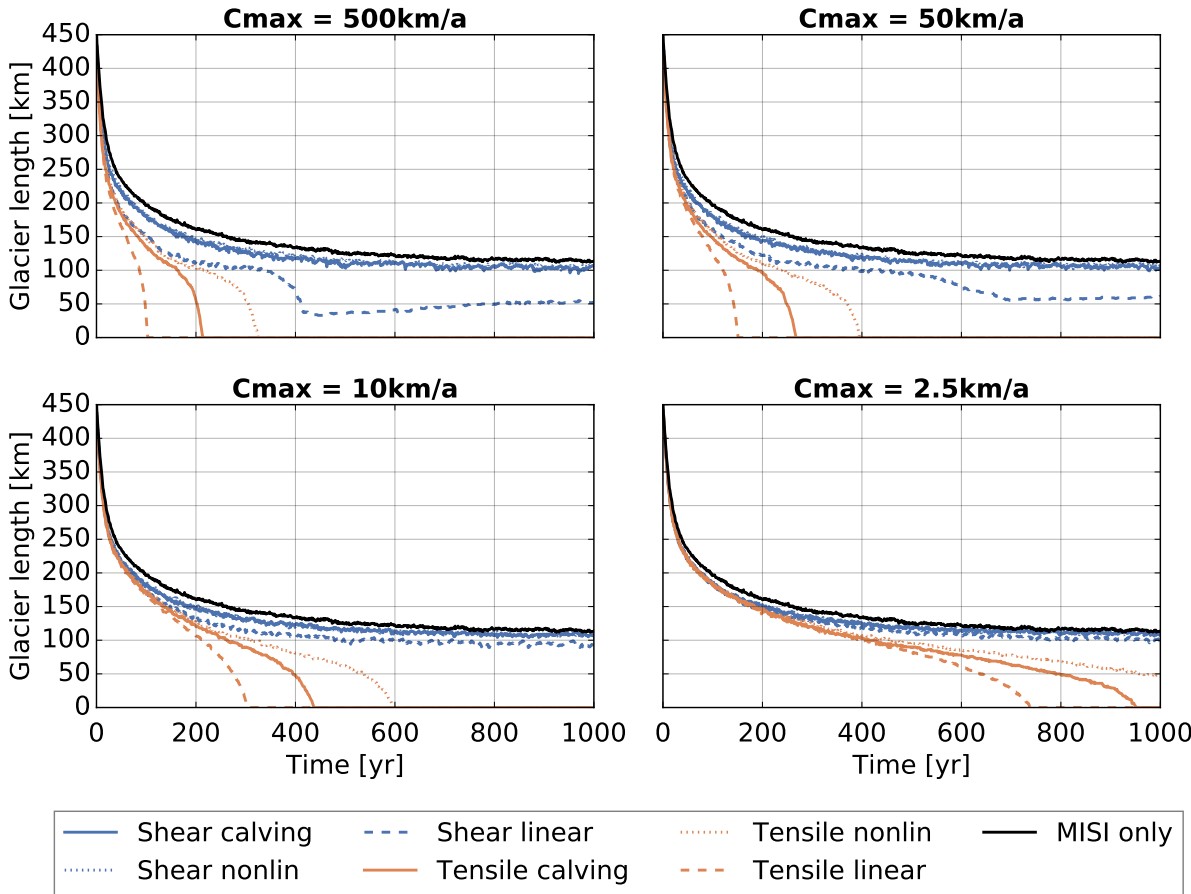

**Figure 7.** Glacier length timeseries. Upper left panel shows runs with an upper limit of $C_{max} = 500\,\text{km/a}$ which is essentially equivalent to the unbuttressed calving rates. Then we have decreasing upper limits and consequently the glacier retreat slows down.

## 5.2 An adaptive upper limit on calving rates

Assuming that mélange equilibration is faster than glacier retreat, the upper bound $C_{max}$ can be calculated as a function of mélange length $L_{em}$. This is further justified by the discussion in section 3.

Here we assume that the position of the embayment exit remains fixed, so that the mélange length grows with the same rate

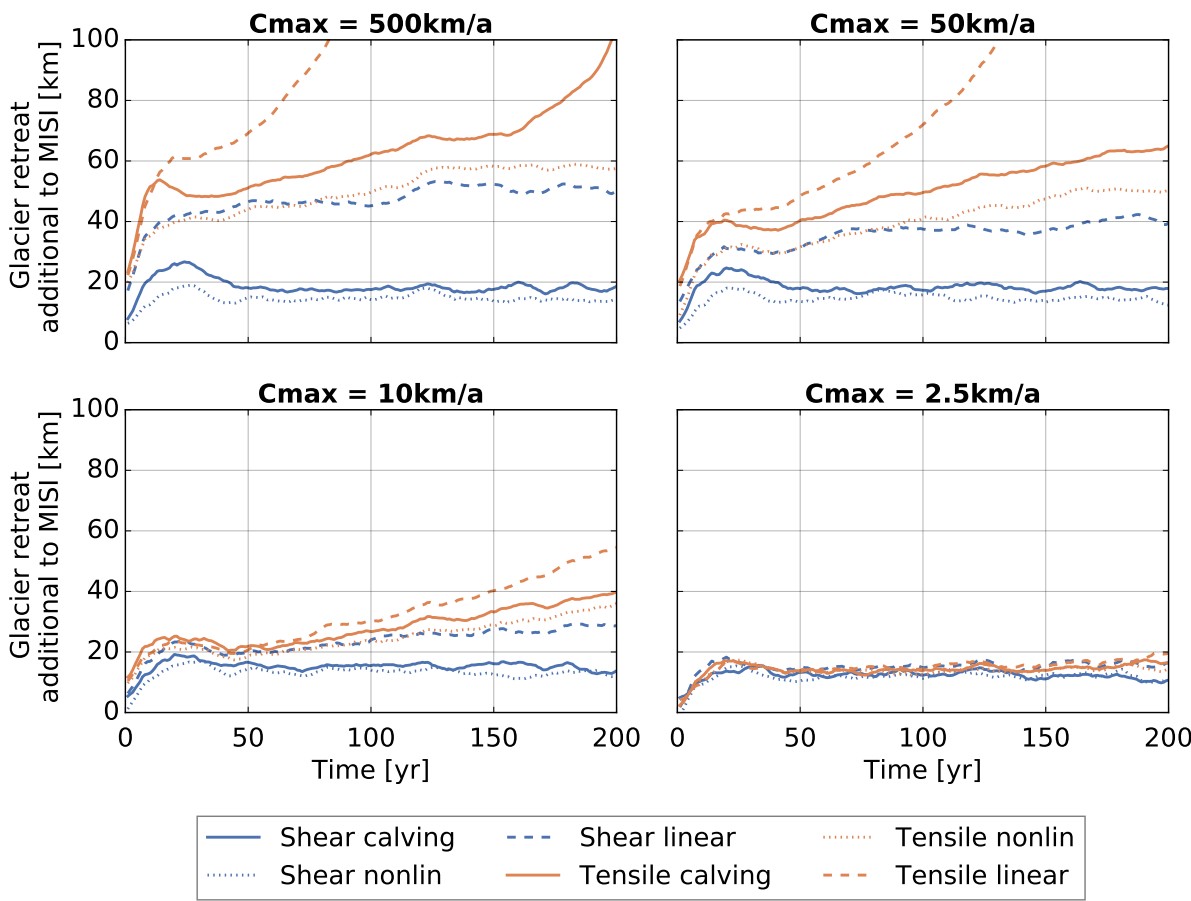

**Figure 8.** Timeseries of glacier retreat additional to the MISI retreat, i.e. retreat caused by calving.

with which the glacier retreats. We assume an initial upper bound $C_{max0} = [2.5, 10.0, 50.0, 500.0]$ km/a at $t = 0$, and update $C_{max}$ each simulation year. We perform the same experiments as described above.

This adaptive approach leads to much smaller calving rates and slows down the glacier retreat significantly (compare fig. 9 to fig. 7). In the case with $C_{max0} = 10$ km/a and $C_{max0} = 2.5$ km/a, the adaptive approach prevents the complete loss of ice. Due to the increase in embayment length, the upper bound in calving rate is reduced to down to $30\%$ of its original value (see fig. 10).

## 6   Conclusions

We considered mélange buttressing of calving glaciers as a complement to previously derived calving relations. These calving relations can lead to unrealistically large calving rates. This is a problem with the calving relations and should be further investigated. Backed by evidence for mélange buttressing in observations and numerical simulations we propose that mélange

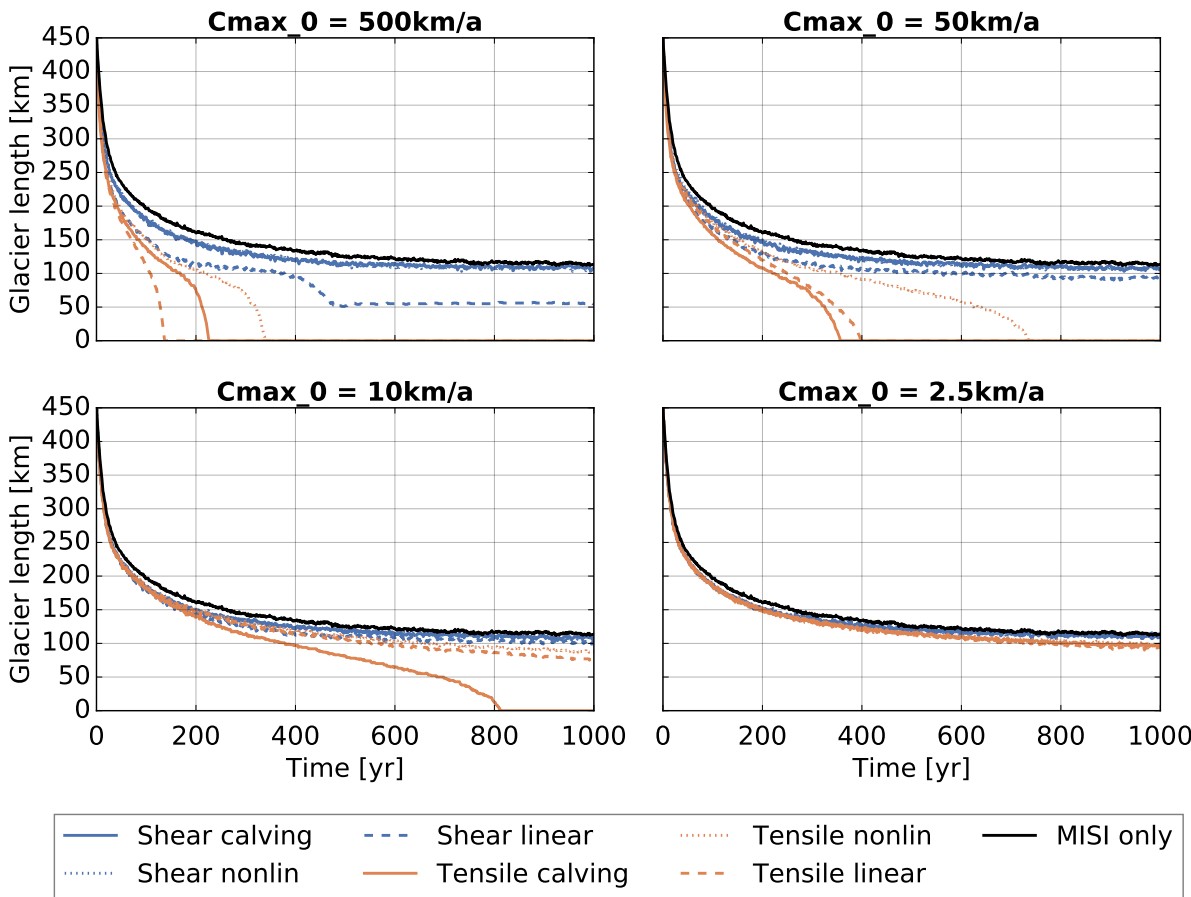

**Figure 9.** Glacier length timeseries with an adaptive calving limit.

buttressing may be one mechanism that prevents calving rates from growing too large. The approach here is to provide an equation that uses simple and transparent assumptions to yield a non-trivial relation.

The central assumption is that the reduction in calving rates is linear with mélange thickness. Other important factors determining the mélange buttressing are the strength of the sea ice bonding the icebergs together (Robel, 2017), and possibly also

5  iceberg size distribution. The continuum rheology model (Amundson and Burton, 2018) adapted here agrees with discrete models (Burton et al., 2018; Robel, 2017) that mélange buttressing increases with the length to width ratio and that is also a feature found here in eq. 8.

The buttressing is described in form of a reduced calving rate which is a functional of the maximum calving rate as it is derived for the ice front without melange buttressing. First, we assumed that calving rates decrease linearly with the mélange

10  thickness. Secondly, we assume a steady state between mélange production through calving and mélange loss through melting and exit from the embayment. This implies a fixed calving front position. Using these two assumptions, we derived a mélange buttressed calving rate, eq. 7, that is linear for small calving rates and converges to an upper limit $C_{max}$, which depends on the

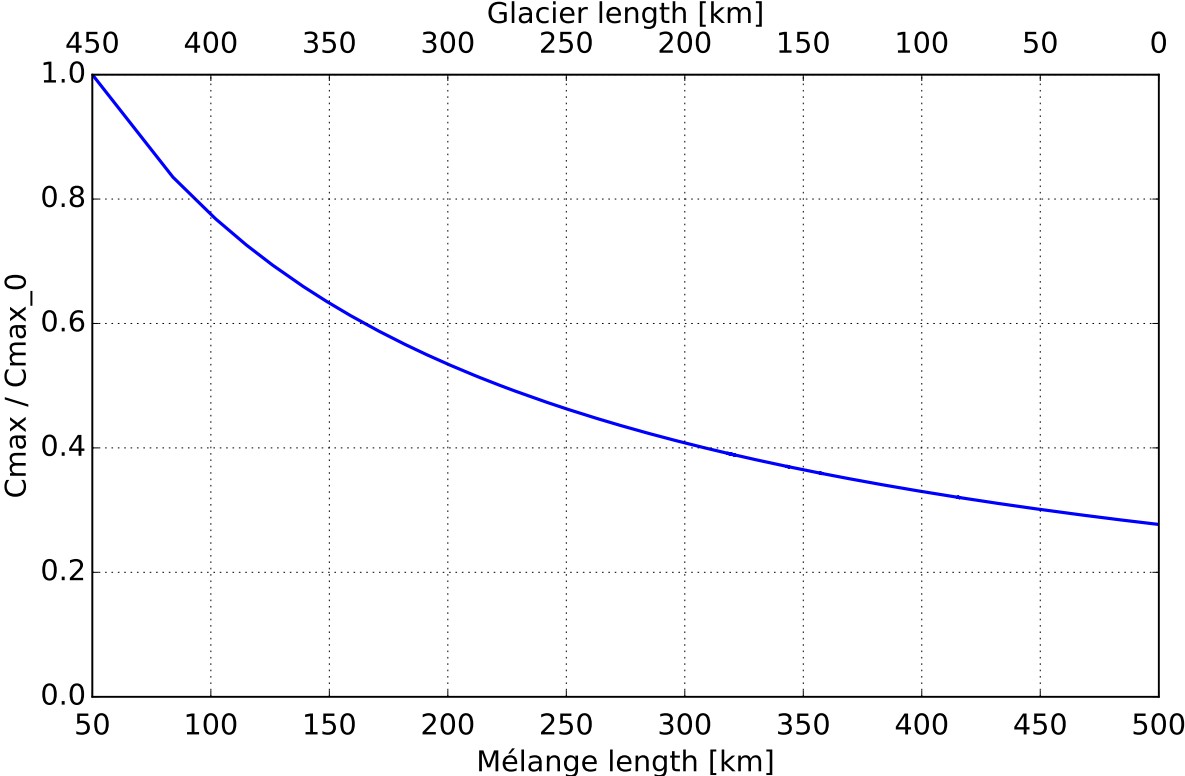

**Figure 10.** Reduction of the upper limit on calving rates as a function of mélange length and glacier length.

embayment geometry, mélange flow properties and the embayment exit velocity.

We also went beyond the steady-state solution of mélange buttressing and considered an evolving mélange geometry. We found that mélange equilibration is faster than glacier retreat, which justifies the use of an adaptive approach in which the upper limit $C_{max}$ is dependent on the mélange geometry.

This framework can be applied to any calving parametrization that gives a calving rate rather than the position of the calving front. We investigated its application to a tensile-failure based calving rate and to a shear-failure based calving rate. For small calving rates, the differences between the parametrizations persist in the buttressed case. However, large calving rates converge to the upper limit and the choice of calving parametrization becomes less important. This suggest that it is possible to simplify

10 the calving parametrizations further, but we show that the simplifications differ for small calving rates and those differences persist.

We illustrated this with a simulation of an idealized glacier. Choice of calving parametrization and choice of upper limit determine the retreat velocity. Following the adaptive approach, glacier retreat leads to a larger embayment and hence larger

mélange buttressing and smaller calving rates.

Embayment geometry plays an important role in determining how susceptible glaciers facing similar ocean conditions are to rapid ice retreat: Pine Island Glacier and Thwaites Glacier in West Antarctica face similar ocean conditions in the Amundsen Sea, where the warming ocean (Shepherd et al., 2004, 2018a) leads to the retreat and rifting of their buttressing ice shelves (Jeong et al., 2016; Milillo et al., 2019), and might be susceptible to both MISI and MICI. Pine Island terminates in an embayment about $45\,\text{km}$ wide, currently filled by an ice shelf of roughly $60\,\text{km}$ length. The upper part of the glacier lies in a straight narrow valley with a width of about $35\,\text{km}$ (distances measured on topography and ice thickness maps provided by Fretwell et al. (2013)). If Pine Island glacier lost its current shelf, it would have a long and narrow embayment holding the ice mélange and would therefore experience strong mélange buttressing. In contrast, Thwaites glacier is more than $70\,\text{km}$ wide and its ice shelf spreads into the open ocean. It has currently no embayment all and once it retreats, it lies in a wide basin that can provide little mélange buttressing. Hence, Thwaites glacier has a much larger potential for large calving rates and runaway ice retreat (MICI) than Pine Island glacier.

Ocean temperatures off the coast of Antarctica are mostly sub-zero with $0.5-0.6°\text{C}$ warming expected until 2200, while the ocean temperatures off the coast of Greenland are sub-zero in the north but up to $4°\text{C}$ in the south with an expected $1.7-2.0°\text{C}$ warming until 2200 (Yin et al., 2011). This leads to increased mélange melting in Greenland compared to Antarctica and therefore higher upper limits on calving rates in Greenland glaciers that have geometries comparable to Antarctic glaciers.
Future ocean warming and intrusion of warm ocean water under the ice mélange increases melting rates and the upper limit on calving rates. This could be another mechanism by which ocean warming increases calving rates.

The concept of cliff calving and a cliff calving instability is not without criticism. According to Clerc et al. (2019), the lower part of the glacier terminus where shear failure is assumed to occur (Bassis and Walker, 2011; Schlemm and Levermann, 2019) is actually in a regime of thermal softening with a much higher critical stress and thus remains stable for large ice thicknesses. Tensile failure may occur in the shallow upper part of the cliff and initiate failure in the lower part of the cliff (Parizek et al., 2019). The critical subaerial cliff height at which failure occurs depends on the timescale of the ice shelf collapse: for collapse times larger than 1 day, the critical cliff height lies between $(170-700\,\text{m})$ (Clerc et al., 2019).

The mélange buttressing model proposed here does not depend on the specific calving mechanism and it is not comprehensive especially since it is not derived from first principles but from a macroscopic perspective. The advantage of the equation proposed here is the very limited number of parameters.

## Appendix A: Mélange thickness gradient

In sec. 2, the mélange thickness was assumed to thin linearly along the embayment length with $d_{cf} = bL_{em}d_{ex}$. Amundson and Burton (2018) give an implicit exponential relation for the mélange thickness:

$$d_{cf} = d_{ex}\exp\left(\mu_0\frac{L_{em}}{W} + \frac{d_{cf} - d_{ex}}{2d_{cf}}\right) \tag{A1}$$

where $\mu_0$ is the coefficient of internal friction of the mélange and ranges from about $0.1$ to larger than $1$. The embayment width, $W$, is assumed to be constant along the embayment in Amundson and Burton (2018), here we can replace it with the average embayment width. In a linear approximation, eq. A1 becomes

$$d_{cf} = d_{ex}\left(1 + \mu_0\frac{L_{em}}{W} + \frac{d_{cf} - d_{ex}}{2d_{cf}}\right) \tag{A2}$$

This equation has one physical solution for $d_{cf}$:

$$d_{cf} = d_{ex} \cdot \frac{1}{4}\left(3 + 2\mu_0\frac{L_{em}}{W} + \sqrt{1 + 12\mu_0\frac{L_{em}}{W} + 4\left(\mu_0\frac{L_{em}}{W}\right)^2}\right) \approx \beta d_{ex} \tag{A3}$$

The parameter $\beta$ can be linearized to take the form given in eq. 4, where the parameters $b_0$ and $b_1$ are determined by the way of obtaining the linear approximation: Completing the square under the squareroot gives the asymptotic upper limit with $b_0 = 1.5$, $b_1 = 1.0$. Taylor expansion can be used to get a more accurate approximation around a specific value of $\mu_0 L/W$: expansion around $\mu_0 L/W = 0.5$ gives $b_0 = 1.11$, $b_1 = 1.21$ while expansion around $\mu_0 L/W = 1.0$ gives $b_0 = 1.17$, $b_1 = 1.11$. The choice of linearisation parameters $b_0$ and $b_1$ should depend on the expected range of values for $\mu_0 L/W$. Fig A1 shows that each of the linear approximations given in the text overestimates $\beta$ slightly but that it is possible to achieve a small error ($< 5\%$) over a rather large range of values for $L/W$.

*Author contributions.* Both authors conceived the study and analysed the data. T.S. developed the basic equations, carried out the experiments and wrote the manuscript. A.L. contributed to the writing of the manuscript.

*Competing interests.* No competing interests

*Acknowledgements.* T.S. was funded by a doctoral scholarship of the H. Boell foundation.

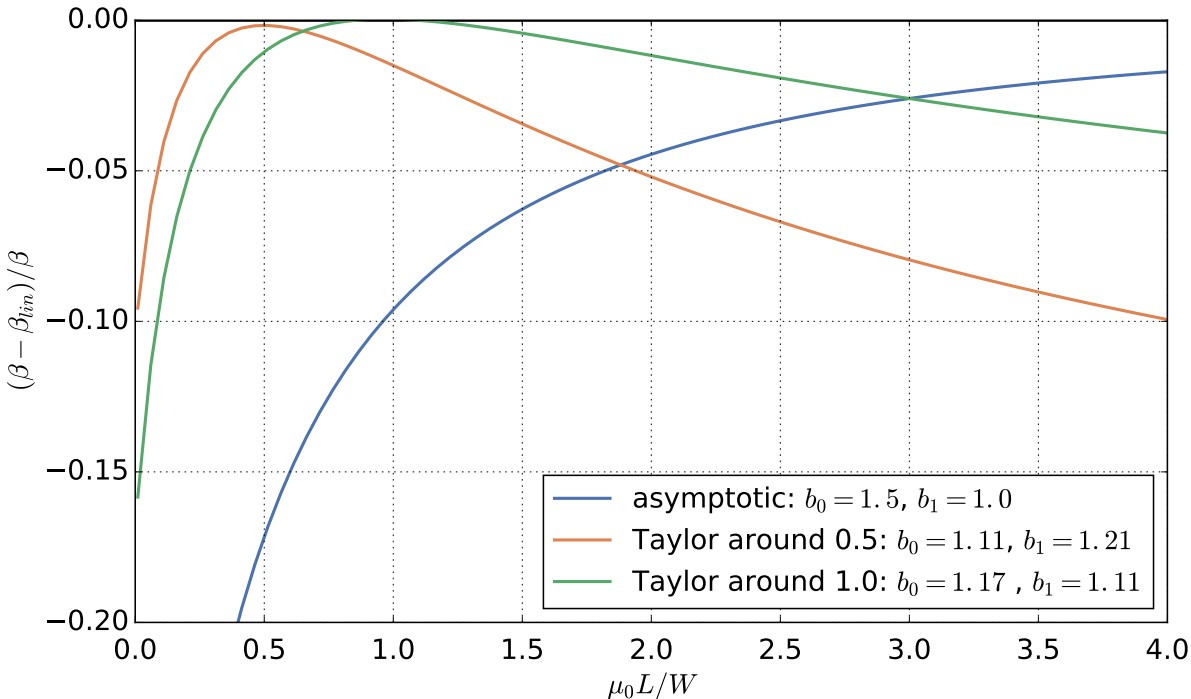

**Figure A1.** The relative difference between $\beta$ given by eq. A3 and different linear approximations of $\beta$.

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
