# Peer review of "A simple parametrization of mélange buttressing for calving glaciers"

_The Cryosphere, 2020_

## Referee Comment (RC1) · Anonymous Referee #1 · 7 Mar 2020

**SUMMARY**

In this study the authors develop a simple parameterization of ice mélange in order to test the idea that there exists a negative feedback loop between iceberg production and buttressing forces from ice mélange. Although I think there is great value in developing simple parameterizations like this, both for understanding basic system behavior and for implementation in long time- and length-scale ice sheet models, I have several concerns about the proposed model. At a minimum, I think the model needs additional explanation and justification. Further model development may also be necessary to make the results more robust.

First, the model is based on an assumed relationship between ice mélange thickness and the "buttressed" calving rate (i.e., the reduction in calving that occurs when ice mélange is present). Although I have some concerns about the exact form of this relationship, it seems like a pretty reasonable starting point since the resistive stress depends on the ice mélange thickness (e.g., see Amundson and Burton, 2018).

Second, the authors use mass continuity to come up with an expression for the ice mélange thickness at the calving face. In doing so, they assume assume a linear thinning rate (if I understand correctly) along the length of the ice mélange, which is probably not a bad assumption but is a bit ad hoc. Note that thickness profiles have been plotted in at least two publications: Amundson and Burton (2018) and Xie et al. (2019). The authors also assume that the ice mélange volume is in steady-state, and then apply their model to non-steady-state situations. That seems dangerous, especially without further justification. I don't understand the consequences of that assumption, which the authors also don't address. In particular, the parameter $a$ is treated as a constant, but it depends on the width of the calving face, the width of the end of the ice mélange, the length of the ice mélange, the velocity of icebergs at the end of the ice mélange, and some unknown flow parameterization $b$. Most or all of these could change with time as the glacier terminus advances/retreats through a fjord and the ice mélange geometry evolves.

As a result of these concerns, I'm not sure how much faith to put in the model results. Essentially, the authors started off with an assumption that there is a negative feedback loop between calving and ice mélange buttressing, and then demonstrated that their model produces a negative feedback loop. This also makes the title feel misleading. I think a more effective approach would be to ask "If ice mélange produces a negative feedback loop with calving, what properties must it have in order to appreciably affect tidewater glacier retreat?"

**SPECIFIC COMMENTS**

- P1, L22: Most studies also neglect the impact of iceberg meltwater on ocean heat transport.

- P3, L6: Amundson and Burton (2018) arrive at a similar result using a very different (continuum mechanics) approach to modeling ice mélange.

- P4, L25: This equation is ad hoc and, as written, not entirely consist with observations. Why does the ice mélange thickness have to equal the terminus thickness to prevent calving from occurring? In general, ice mélange thickness is considerably less than the terminus thickness. Note also that here $d$ is used to refer to the effective ice mélange thickness, but later $d_{cf}$ is used to refer to the thickness at the calving front and substituted into this equation, which is confusing.

- P6, L9-13: This is unnecessarily wordy. You could just write that conservation of mass dictates that $dV/dt = \ldots$, and then explain each of the three terms.

- P6, L11-12: The overall rate of mélange volume "change"?

- P6, L13: This equation shouldn't be set to 0, because its not until the next equation that you assume steady-state.

- P6, L18: How does $b$ parameterize the flow? Are you just suggesting that this is something that could be taken from observations? Please elaborate.

- This is an upper limit, assuming steady-state geometry and flow...

- P7, L8-9: "as also suggested by previous studies."

- P13, L6-7: Please elaborate on what sort of observations could be made. How do you move forward from using steady-state assumptions?

**REFERENCES**

Amundson, J.M. and J.C. Burton, 2018. Quasi-static granular flow of ice mélange, J. Geophys. Res. Earth Surf., 123, 2243–2257, `https://doi.org/10.1029/2018JF004685`.

Xie, S., T.H. Dixon, D.M. Holland, D. Voytenko, and I. Vaňková, 2019. Rapid iceberg calving following removal of tightly packed pro-glacial mélange, Nature Comm., 10, 3250, `https://doi.org/10.1038/s41467-019-10908-4`.

---

## Author Comment (AC1) · 12 Mar 2020

We thank the reviewer for the thoughtful comments which will improve our manuscript greatly. In a revised manuscript will incorporate them in the following manner:

- Mélange thickness to prevent calving:
  We agree with the reviewer that mélange thickness is usually much less than glacier thickness and this is sufficient to prevent calving. Consequently it is probably more realistic to assume calving is inhibited when mélange thickness has reached some fraction $h = cH$, $0 < c < 1$ of the ice thickness. This introduces the factor $c$ into equation (4) without changing the result qualitatively.

- Steady-state assumption:

[Figure]

We also agree with the reviewer that applying a steady-state solution to a scenario with calving front retreat is difficult. We will address this by repeating the idealized scenario experiments of section 4 with a time-dependent upper bound $C_{max} = a^{-1}$. The argument for applying such a quasi-equilibrium approach (without explicitly accounting for the time derivatives for example in the melange volume) will be given in a revised manuscript along the following line: Glacier retreat happens on a timescale of months to years while mélange equilibration is expected to happen much faster. This means that each retreat of the calving front leads to an updated mélange geometry which gives an updated $C_{max}$. In particular, keeping the position where mélange exits into the ocean fixed, the length of the mélange, $L_{em}$, increases when the calving front retreats, leading to smaller upper bounds $C_{max}$, i.e. slower glacier retreat.
* * *

---

## Referee Comment (RC2) · Douglas Benn (Referee) · 25 May 2020

I like this paper. It makes a significant and important contribution to the emerging understanding of how iceberg mélange can limit calving losses, especially with respect to the important issue of marine ice sheet instability. The model shows that, under certain assumptions, the evacuation of mélange places an upper limit on calving losses. This is an important idea and indeed is probably the only plausible process that might limit rates of ice loss from parts of West Antarctica if fringing ice shelves are lost and ice cliffs retreat into deepening water.

However, the model is illustrative rather than predictive. Both the calving laws and the mélange flux equations are untested against observations and cannot be used in their

current form to predict actual ice sheet evolution. Many of the functions are chosen for convenience rather than known fidelity. This of course is a useful modelling strategy, and the untested nature of key equations does not diminish the importance of the paper or undermine the analysis. The paper yields several important insights and is likely to be well cited and influential. On the other hand, the model cannot necessarily be applied to real-world situations simply via calibrating model parameters, and there may also be structural issues.

For example, backstress from mélange does not always come from fjord walls, but also via grounding of bergs on the sea bed. In some circumstances, the effect of mélange buttressing may be better posed as a force balance and ice-margin position problem, rather than a rate problem as developed here. But I am not suggesting that the analysis needs to be augmented or changed in any way, only that in a couple of places statements should be added to acknowledge the limitations. I suggest:

1. In the abstract, line 5: delete 'but robust'. The robustness of a model can only be evaluated with respect to its predictive or diagnostic power, which is not addressed in this paper.

2. Add a sentence or two to Section 5 to mention the possible limitations of model structure.

3. In p 15, line 20, the authors state that an advantage of their model is that it employs a limited number of parameters that can be calibrated against observations. But this may not be possible. I suggest simply deleting this sentence.

The text is very clearly written, and I only spotted a few typos.

Abstract, line 2: 'calf' should be 'calve'

P 7, line 15: delete 'a'

P 7, line 16: 'instable' should be 'unstable'

The authors should be congratulated for their original and very useful contribution.

---

## Referee Comment (RC3) · Anonymous Referee #3 · 1 Jun 2020

This paper attempts to set a physically motivated bound on calving rates (this is clear from the paper but perhaps not from the abstract). This is an important task and the authors take a reasonable approach towards this goal.

I have a number of comments which I think should be addressed before this paper is accepted:

Numerical experiment

Because the boundary condition on the sides of the channel are periodic, in this setup any potentially formed ice shelf would be unconstrained and therefore incapable of providing ice shelf buttressing. To some extent ice melange can be though of as a weak ice shelf with different rheology, and therefore melange buttressing will also be absent in a

setup that does not allow ice shelf buttressing. I find the fact that a melange buttressing parameterization is tested in a setup that does not allow for ice shelf/melange buttressing to begin with inconsistent. Using no slip boundary conditions on the side walls would solve this inconsistency. For the no slip wall case then, the effective melange buttressing can be diagnosed from the model and compared with observed and modeled values of melange strength.

Simplified calving relations

Section 3.4 doesn't make much sense. Calving relations are simplified by a fitting a function to a region generated by considering different water depths and freeboards. This simplified relation is then used in the numerical simulation. Because the water depth is known exactly in a given setup (the numerical experiment) an exact calving relation should be used directly, rather than a fit to the range of values generated from multiple water thicknesses. If this were not computationally feasible, linearization locally using Taylor expansion should be used, not an arbitrary global line fit.

Melange properties

The authors ignore the granular character of the melange. Because melange is a sea ice/ice berg mixture, it is its concentration that has bigger impact on its strength than its thickness. Thickness becomes relevant only when the concentration is close to 1. Yet, in this paper it is thickness that is the key variable in deriving the bound on calving rate. It should be either stated that concentration is assumed to be 1, which is unrealistic, or the melange concentration should be taken into account, perhaps by elaborating on the relationship between melange thickness and melange effective thickness

The authors use the terms melange thickness and melange effective thickness interchangeably, however these are not the same. This has an effect on the mass conservation in equation 2. Because melange thickness is not melange effective thickness, the calving rate does not equal the rate of melange formation at the calving front. This needs to be addressed/corrected.

Melange flow and material properties are all lumped into one parameter b, it should be justified what the reasonable range of b is. There should also be a way to translate this parameter b to melange strength (under some assumptions) so that there is a clear way to evaluate the parameterization in the future when more observations become available. Also, as b is likely to be bounded because realistic melange has a finite maximum strength; this has implications for constraining the value of Cmax for a given embayment geometry.

Minor:

*Forcing in the numerical experiments is unclear - why is the ice shelf removed throughout the simulations, rather than just at the initial time of each experiment?

*Figures not well referenced through the text - there is a lot of statements floating around and it is unclear if they are based on a figure or equation or some previous work.

More comments and suggestions are included in the attached pdf.

Please also note the supplement to this comment:
https://www.the-cryosphere-discuss.net/tc-2020-50/tc-2020-50-RC3-supplement.pdf

—————————————————

[Figure]

**Supplement:**

[revised manuscript text omitted]

---

## Author Comment (AC2) · 17 Jul 2020

**Authors' reply**

T. Schlemm, A. Levermann

July 17, 2020

We thank all referees for their helpful comments that stipulated further investigations into the gradient of the mélange thickness and mélange buttressing beyond the steady state. We improve the model presented in the manuscript by showing that it can also be solved without the assumption of a fixed mélange geometry. For application in glacier retreat modelling, an adaptive approach can be used in which the upper bound on calving rates is updated when the mélange geometry changes. We first reply to the referees individually and then present the new work inspired by the referees' comments in the appendix.

**1 Anonymous Referee 1**

**Main comments**

The authors also assume that the ice mélange volume is in steady- state, and then apply their model to non-steady-state situations. That seems dangerous, especially without further justification. I don't understand the consequences of that assumption, which the authors also don't address. In particular, the parameter a is treated as a constant, but it depends on the width of the calving face, the width of the end of the ice mélange, the length of the ice mélange, the velocity of icebergs at the end of the ice mélange, and some unknown flow parameterization b. Most or all of these could change with time as the glacier terminus advances/retreats through a fjord and the ice mélange geometry evolves.

**Response:** The mélange buttressing model can be modified to allow free evolution of mélange geometry and this justifies using an adaptive approach in glacier retreat simulations, in which the upper bound  $C_{max}$  is calculated after each time step to account for the change in mélange geometry (see sections B and C).

Essentially, the authors started off with an assumption that there is a negative feedback loop between calving and ice mélange buttressing, and then demonstrated that their model produces a negative feedback loop. This also makes the title feel misleading. I think a more effective approach would be to ask "If ice mélange produces a negative feedback loop with calving, what properties must it have in order to appreciably affect tidewater glacier retreat?"

**Response:** We argue that there are good reasons to assume a negative feedback loop between mélange thickness and calving rates. We do not claim to prove the existence of this feedback loop, rather we show that this negative feedback loop causes an upper bound on the calving rate which depends on the embayment gemoetry (width and length) and on mélange properties (internal friction and exit velocity). Thus we do not prove that mélange buttressing exists (other papers show good evidence for this assumption) but rather show how it may effect calving rates. This means we essentially present a model for mélange buttressing, and that's why we find the title of the manuscript appropriate and ask the referee to allow it.

**Minor comments**

P1, L22: Most studies also neglect the impact of iceberg meltwater on ocean heat transport. **Response:** Yes, that's true. We approach this from an ice-sheet-modelling approach rather than an ocean-modelling view-point, so we do not consider iceberg meltwater, either. P3, L6: Amundson and Burton (2018) arrive at a similar result using a very different (continuum mechanics) approach to modeling ice mélange. **Response:** Thank you for pointing this out.

P4, L25: This equation is ad hoc and, as written, not entirely consist with observations. Why does the ice mélange thickness have to equal the terminus thickness to prevent calving from occurring? In general, ice mélange thickness is considerably less than the terminus thickness. Note also that here d is used to refer to the effective ice mélange thickness, but later d cf is used to refer to the thickness at the calving front and substituted into this equation, which is confusing.

**Response:** As stated in our previous authors' reply, we assume now that calving is inhibited when mélange thickness has reached some fraction  $h = \gamma H$ ,  $0

Figure 1: The relative difference between  $\beta$  given by eq. 3 and different linear approximations of  $\beta$ .

where L(t) is the distance between the the embayment exit and the calving front, W(x) the width of the embayment at a distance x from the embayment exit and d(x,t) is the mélange thickness. This expression is equal to the sum of mélange production and loss terms given in eq. 2 in the manuscript. By applying the Leibniz integral rule to the volume integral of eq. 5 as well as rewriting the mélange production and loss terms as functions of time and calving front position, eq. 2 in the manuscript becomes

$$W(L(t))H(L(t))C(t) - W(0)d(0,t)u_{ex} - m \int_{0}^{L(t)} dx W(x)$$

=  $\frac{d}{dt}(L(t)) \cdot W(L(t)) \beta(L(t))d(0,t) + \frac{d}{dt} (\beta(L(t))d(0,t)) \cdot \int_{0}^{L(t)} dx W(x)$  (6)

where the first three terms on the left hand side are the mélange production through calving, the mélange loss at the embayment exit and the mélange melting, respectively, and the left hand side is the rewritten volume integral. This differential equation for d(0,t) can be solved if the embayment geometry W(x) as well as ice thickness at the calving front H(L(t)) are known, the calving rate C(t) is given by

$$C(t) = \left(1 - \frac{\beta(L(t))d(0,t)}{\gamma H(L(t))}\right)C^*$$
(7)

and the change rate of the embayment length, L(t), is given by

$$\frac{\mathrm{d}}{\mathrm{d}t}L(t) = C(t) - u_{cf}(t) \tag{8}$$

where the ice flow velocity at the calving front,  $u_{cf}(t)$ , depends on the bed topography and the ice dynamics.

We will now consider an idealized setup with constant ice thickness, H(x) = H, as well as constant embayment width, W(x) = W, while neglecting ice flow by setting  $u_{cf} = 0$ . Eqs. 6 - 8 are solved numerically for the parameter values H = 1000 m, W = 10 km,  $\mu = 0.3$ ,  $\gamma = 0.2$ ,  $C^* = 3 \text{ km/a}$ ,  $u_{ex} = 100 \text{ km/a}$ ,  $b_0 = 1.11$ ,  $b_1 = 1.21$ , and the initial conditions L(0) = 10 km and d(0) = 10 m. We consider a scenario without mélange melting, m = 0, and a scenario with mélange melting, where the melt rate is set to m = 10 m/a (see fig. 2). In the scenario without melting, mélange length and thickness at the calving front increase, while mélange thickness at the embayment exit and buttressed calving rate decrease. If melting of mélange is considered, the mélange thickness at the calving front increases initially, and then decreases until the embayment is mélange-free, since the volume of mélange melted increases with mélange area. A comparison between these solutions, where the mélange geometry is free to evolve, and the steady-state solution obtained by plugging the mélange length, L(t), into eq. 4 and 6 in the manuscript, respectively, (see bottom panels of fig. 2) shows good aggreement. This is a justification for the adaptive approach discussed in next section.

---

## Author Response (AR1)

**Authors' reply**

T. Schlemm, A. Levermann

August 26, 2020

We thank all referees for their helpful comments that stipulated further investigations into the gradient of the mélange thickness and mélange buttressing beyond the steady state. We improve the model presented in the manuscript by showing that it can also be solved without the assumption of a fixed mélange geometry. For application in glacier retreat modelling, an adaptive approach can be used in which the upper bound on calving rates is updated when the mélange geometry changes.

**1 Anonymous Referee 1**

**Main comments**

*The authors also assume that the ice mélange volume is in steady- state, and then apply their model to non-steady-state situations. That seems dangerous, especially without further justification. I don't understand the consequences of that assumption, which the authors also don't address. In particular, the parameter a is treated as a constant, but it depends on the width of the calving face, the width of the end of the ice mélange, the length of the ice mélange, the velocity of icebergs at the end of the ice mélange, and some unknown flow parameterization b. Most or all of these could change with time as the glacier terminus advances/retreats through a fjord and the ice mélange geometry evolves.*
**Response:** The mélange buttressing model has been be modified to allow free evolution of mélange geometry and this justifies using an adaptive approach in glacier retreat simulations, in which the upper bound $C_{max}$ is calculated after each time step to account for the change in mélange geometry (see sections 3 and 5.2 in the revised manuscript).

*Essentially, the authors started off with an assumption that there is a negative feedback loop between calving and ice mélange buttressing, and then demonstrated that their model produces a negative feedback loop. This also makes the title feel misleading. I think a more effective approach would be to ask "If ice mélange produces a negative feedback loop with calving, what properties must it have in order to appreciably affect tidewater glacier retreat?"*
**Response:** We argue that there are good reasons to assume a negative feedback loop between mélange thickness and calving rates. We do not claim to prove the existence of this feedback loop, rather we show that this negative feedback loop causes an upper bound on the calving rate which depends on the embayment gemoetry (width and length) and on mélange properties (internal friction and exit velocity). Thus we do not prove that mélange buttressing exists (other papers show good evidence for this assumption) but rather show how it may effect calving rates. This means we essentially present a model for mélange buttressing, and that's why we find the title of the manuscript appropriate and ask the referee to allow it.

**Minor comments**

*P1, L22: Most studies also neglect the impact of iceberg meltwater on ocean heat transport.*
**Response:** Yes, that's true. We approach this from an ice-sheet-modelling approach rather than an ocean-modelling view-point, so we do not consider iceberg meltwater, either.

*P3, L6: Amundson and Burton (2018) arrive at a similar result using a very different (continuum mechanics) approach to modeling ice mélange.*
**Response:** Thank you for pointing this out, the reference has been included (P3, L7)

*P4, L25: This equation is ad hoc and, as written, not entirely consist with observations. Why does the ice mélange thickness have to equal the terminus thickness to prevent calving from occurring? In general, ice mélange thickness is considerably less than the terminus thickness. Note also that here d is used to refer to the effective ice mélange thickness, but later d cf is used to refer to the thickness at the calving front and substituted into this equation, which is confusing.*
**Response:** As stated in our previous authors' reply, we assume now that calving is inhibited when mélange thickness has reached some fraction $h = \gamma H$, $0 < \gamma < 1$ of the ice thickness. This introduces the factor $\gamma$ into equation (1) and equation (6) without changing the result qualitatively.

*P6, L9-13: This is unnecessarily wordy. You could just write that conservation of mass dictates that dV /dt = . . ., and then explain each of the three terms.*
**Response:** We corrected this (P6, L10...)

*P6, L11-12: The overall rate of mélange volume "change"?*
**Response:** Yes, this was corrected. (P6, L9)

*P6, L13: This equation shouldn't be set to 0, because its not until the next equation that you assume steady-state.*
**Response:** Thank you for noticing this, we corrected it. (P6, L13)

*P6, L18: How does b parameterize the flow? Are you just suggesting that this is something that could be taken from observations? Please elaborate.*
**Response:** The parameter b giving the mélange gradient along the embayment is now determined by linearizing the implicit exponential equation given in Amundson, Burton (2018). It then depends on the coefficient of internal friction of the mélange $\mu_0$ (see appendix in the revised manuscript).

*P7, L8-9: "as also suggested by previous studies."*
**Response:** This was included. (P8, L4-5)

*P13, L6-7: Please elaborate on what sort of observations could be made. How do you move forward from using steady-state assumptions?*
**Response:** As section 3 in the revised mansucript shows, it is justified to use the steady-state model for glacier retreat if an adaptive upper bound on calving rates is used. Observations could further constrain the internal friction of mélange and the velocity of mélange exiting the embayment.

**2  Douglas Benn**

We thank Doug Benn for his thoughtful review and the positive feedback. The minor comments have been taken into account and corrections made.

**3  Anonymous Referee 3**

**Major comments**

*Numerical experiment: Because the boundary condition on the sides of the channel are periodic, in this setup any potentially formed ice shelf would be unconstrained and therefore incapable of providing ice shelf buttressing. To some extent ice melange can be though of as a weak ice shelf with different rheology, and therefore melange buttressing will also be absent in a setup that does not allow ice shelf buttressing. I find the fact that a melange buttressing parameterization is tested in a setup that does*

*not allow for ice shelf/melange buttressing to begin with inconsistent. Using no slip boundary conditions on the side walls would solve this inconsistency. For the no slip wall case then, the effective melange buttressing can be diagnosed from the model and compared with observed and modeled values of melange strength.*

**Response:** This is a misunderstanding. The setup has rocky fjord walls and where the bedrock wall is below sea level, there is grounded ice resting on it. The Spinup has an ice shelf constrained by these grounded ice walls which is exerting a buttressing force on the glacier. This is why the removal of the ice shelf leads to a rapid glacier retreat already without any calving parametrisation applied (MISI only in fig. 5). This has been clarified in the manuscript. (P14, L25..)

*Simplified calving relations: Section 3.4 doesn't make much sense. Calving relations are simplified by a fitting a function to a region generated by considering different water depths and freeboards. This simplified relation is then used in the numerical simulation. Because the water depth is known exactly in a given setup (the numerical experiment) an exact calving relation should be used directly, rather than a fit to the range of values generated from multiple water thicknesses. If this were not computationally feasible, linearization locally using Taylor expansion should be used, not an arbitrary global line fit.*

**Response:** The purpose of the simplifications is not to replace the full calving parametrisations in numerical simulations, but rather to be illustrative. The combination of a nonlinear calving relation and nonlinear buttressing makes it difficult to isolate the effect of mélange buttressing. The simplifications make the relation a bit clearer. We ask the reviewer to allow this.

*Melange properties: The authors ignore the granular character of the melange. Because melange is a sea ice/ice berg mixture, it is its concentration that has bigger impact on its strength than its thickness. Thickness becomes relevant only when the concentration is close to 1. Yet, in this paper it is thickness that is the key variable in deriving the bound on calving rate. It should be either stated that concentration is assumed to be 1, which is unrealistic, or the melange concentration should be taken into account, perhaps by elaborating on the relationship between melange thickness and melange effective thickness.*

*The authors use the terms melange thickness and melange effective thickness interchangeably, however these are not the same. This has an effect on the mass conservation in equation 2. Because melange thickness is not melange effective thickness, the calving rate does not equal the rate of melange formation at the calving front. This needs to be addressed/corrected.*

*Melange flow and material properties are all lumped into one parameter b, it should be justified what the reasonable range of b is. There should also be a way to translate this parameter b to melange strength (under some assumptions) so that there is a clear way to evaluate the parameterization in the future when more observations become available. Also, as b is likely to be bounded because realistic melange has a finite maximum strength; this has implications for constraining the value of Cmax for a given embayment geometry.*

**Response:** We have chosen to stick with the average mélange thickness rather than using an effective mélange thickness. The parameter b giving the mélange gradient along the embayment is now determined by linearizing the implicit exponential equation given in Amundson, Burton (2018). It then depends on the coefficient of internal friction of the mélange (which ranges from 0.1 to above 1) (see the appendix in the revised mansucript).

**Minor comments**

*Forcing in the numerical experiments is unclear - why is the ice shelf removed throughout the simulations, rather than just at the initial time of each experiment?*

**Response:** PISM tends to regrow shelves very quickly. If floating ice was removed only in the first time step, at least one cell of floating ice would regrow within the first simulation year and form the new glacier terminus. Since the calving parametrisations are applied only to grounded termini, the shelf would not be calved off and continue to grow. In order to prevent this spurious regrowth of a floating tongue, floating ice is removed at every time step.

*Figures not well referenced through the text - there is a lot of statements floating around and it is unclear if they are based on a figure or equation or some previous work.*
**Response:** We have corrected this in a number of places.

*Please also note the supplement to this comment*
**Response:** Comments in the supplement were taken into account and corrections made.

[revised manuscript text omitted]
{\mathrm{d}V}{\mathrm{d}t} = \frac{\mathrm{d}}{\mathrm{d}t} \int\limits_0^{L(t)} \mathrm{d}x \, W(x) \, d(x,t) \tag{9}$$

where $L(t)$ is the distance between the the embayment exit and the calving front, $W(x)$ the width of the embayment at a distance $x$ from the embayment exit, $d(x,t)$ is the mélange thickness and the embayment exit is fixed at $x=0$. This expression

5  is equal to the sum of mélange production and loss terms given in eq. 2. By applying the Leibniz integral rule to the volume integral of eq. 9 as well as rewriting the mélange production and loss terms as functions of time and calving front position, eq. 2 becomes

$$W_L H C - W_0 d_0 u_{ex} - m \int\limits_0^L \mathrm{d}x \, W(x) = W_L \beta d_0 \cdot \frac{\mathrm{d}}{\mathrm{d}t} L + \left( \int\limits_0^L \mathrm{d}x \, W(x) \right) \cdot \frac{\mathrm{d}}{\mathrm{d}t} \left( \beta d_0 \right) \
[revised manuscript text omitted]

---

## Referee Report (RR1)

**SUMMARY**

The authors have addressed each of the comments from my first review. I particularly appreciate the discussion that they have included to show that, in their model, the ice mélange evolves quite quickly, which suggests that it can be treated as being approximately in steady-state. However, the new figure that they added (Fig. 3) raises some concerns that I did not identify previously. In their model, once the ice mélange thins to 0 at the mouth of the embayment it is no longer capable of affecting the calving rate, even if the ice mélange is a couple of hundred kilometers long! I don't see why that should be. Most of the studies that suggest that ice mélange can affect calving are from Jakobshavn Isbræ. There, the ice mélange rarely extends beyond 15–20 km in length, whereas the fjord into which the glacier flows is ∼50 km long. I think the issue is with Equation 4,

$$d_{cf} = \beta d_{ex},$$

which states that the ice mélange thickness at the terminus is proportional to the thickness at the end of the embayment. Once the thickness at the end of the embayment goes to 0, the thickness at the terminus also goes to zero. Essentially the ice mélange is pinned to the end of the embayment, and as a glacier retreats it gets stretched thinner and thinner (although the thinning is offset some by increased iceberg production). This seems to be a pretty serious issue that should be addressed as it affects all of the subsequent interpretation. Why should the ice mélange have to extend to the end of the embayment?

A couple of other general comments:

- Perhaps worth discussing in a few sentences why you adopt a continuum modeling approach for ice mélange, and how this is motivated/justified by attempts in the granular mechanics community to develop continuum rheologies for granular materials.

- This paper seems to be motivated by the observation that some calving parameterizations don't seem to have an upper limit on calving rates and can produce very high and unrealistic calving rates. I think we need to ask if there is something fundamentally wrong with the physics of those calving models. Although I generally like the approach taken here and do feel that ice mélange can reduce calving rates, I would suggest putting less emphasis on trying to fix those models with an "ice mélange bandage".

**SPECIFIC COMMENTS**

P2, L5–17: These two paragraphs are kind of choppy.

P6, L4–9: Some of these variables were already defined on the previous page.

P6, L12: Perhaps say that $m$ is the average melt rate instead of assuming that it is spatially constant?

P7, L4: Suggest "Assuming a viscoplastic rheology and quasi-static flow"

P7, Eq. 4: This equation is where my concerns start. It indicates that the ice mélange thickness at the calving face is zero if there is no thickness at the mouth of the embayment. (See summary comments.)

P7, L16–20: I think this paragraph could more clearly state the implications of this model. Essentially, if the calving rate is low, ice mélange will have little impact on the calving rate because the ice mélange doesn't become expansive enough. Only when the calving rate is high does ice mélange

become important. I'm not sure if that is physically correct—it could be—but at any rate it does seem to be a feature of this model.

P11, L8 & L22: ma$^{-1}$ should be m a$^{-1}$ (milliyears vs. meters per year)

P15, L16–17: This is confusing, since in Section 3 you have argued that the ice mélange quickly adjust its geometry. Why not just use the adaptive approach?

P18, L1: I think you can express the "position-based" calving parameterizations in terms of rates that depend on the thickness gradient. I'm not sure if that is written up anywhere, but my point is that you can probably use this framework for other types of parameterizations than what you have considered here.

---

## Author Response (AR2)

Dear Dr. Nisancioglu,

Thank you very much for handling our review process. The reviewer comments were very constructive and we are confident that we have addressed them fully and that the paper is now ready for publication.

Best wishes,
Anders Levermann and Tanja Schlemm

**Referee 1**

*However, the new figure that they added (Fig. 3) raises some concerns that I did not identify previously. In their model, once the ice mélange thins to 0 at the mouth of the embayment it is no longer capable of affecting the calving rate, even if the ice mélange is a couple of hundred kilometers long! I don't see why that should be. Most of the studies that suggest that ice mélange can affect calving are from Jakobshavn Isbræ. There, the ice mélange rarely extends beyond 15–20 km in length, whereas the fjord into which the glacier flows is ∼50 km long. I think the issue is with Equation 4, $d\_cf = \beta\, d\_ex$ , which states that the ice mélange thickness at the terminus is proportional to the thickness at the end of the embayment. Once the thickness at the end of the embayment goes to 0, the thickness at the terminus also goes to zero. Essentially the ice mélange is pinned to the end of the embayment, and as a glacier retreats it gets stretched thinner and thinner (although the thinning is offset some by increased iceberg production). This seems to be a pretty serious issue that should be addressed as it affects all of the subsequent interpretation. Why should the ice mélange have to extend to the end of the embayment?*

**Reply:** We thank the referee for pointing this issue out. It's true that the mélange does not necessarily need to be pinned to the embayment exit. An additional equation for the mélange length, L, would improve the model greatly, but we cannot find any in the relevant literature.
In section 3, we included the case with a fixed mélange length, L(t) = L, which corresponds to a mélange retreating with the glacier front. In this case the mélange thickness equilibrates quickly and results in a constant upper bound C_max. This corresponds to the idealized channel simulation performed in section 5.1

*Perhaps worth discussing in a few sentences why you adopt a continuum modeling approach for ice mélange, and how this is motivated/justified by attempts in the granular mechanics community to develop continuum rheologies for granular materials.*

**Reply:** We adopt a continuum modeling approach because it is simpler and because it gives an analytical equation for mélange thickness that we can adapt. Continuum and discrete modeling approaches differ in some details, but they agree that mélange buttressing strength increases with the mélange length-to-width ratio – a feature which our parametrization also shows. A few sentences have been added to the discussion.

*This paper seems to be motivated by the observation that some calving parameterizations don't seem to have an upper limit on calving rates and can produce very high and unrealistic calving rates. I think we need to ask if there is something fundamentally wrong with the physics of those calving models. Although I generally like the approach taken here and do feel that ice mélange can reduce calving rates, I would suggest putting less emphasis on trying to fix those models with an "ice mélange bandage"*

**Reply:** Good point. A few sentences have been added to the discussion.

*P2, L5–17: These two paragraphs are kind of choppy.*

**Reply:** Rewritten

*P6, L4–9: Some of these variables were already defined on the previous page.*

**Reply:** corrected

*P6, L12: Perhaps say that m is the average melt rate instead of assuming that it is spatially constant?*

**Reply:** done

*P7, L4: Suggest "Assuming a viscoplastic rheology and quasi-static flow"*

**Reply:** done

*P7, Eq. 4: This equation is where my concerns start. It indicates that the ice mélange thickness at the calving face is zero if there is no thickness at the mouth of the embayment. (See summary comments.)*

**Reply:** The way we correct this is to say that mélange does not need to be pinned to the embayment exit. So this equation relates mélange thickness at the calving front and at the end of the mélange.

*P7, L16–20: I think this paragraph could more clearly state the implications of this model. Essentially, if the calving rate is low, ice mélange will have little impact on the calving rate because the ice mélange doesn't become expansive enough. Only when the calving rate is high does ice mélange become important. I'm not sure if that is physically correct—it could be—but at any rate it does seem to be a feature of this model.*

**Reply:** It depends on the mélange geometry and the resulting Cmax. If the unbuttressed calving rate C* is much smaller than Cmax, then there is indeed little buttressing. However, if both rates are small, but C* is close to Cmax then there is significant buttressing. So it's not the absolute value of the calving rate that determins the strength of the buttressing effect, but its ratio to the upper bound Cmax which depends on the embayment geometry and mélange length.

*P11, L8 & L22: ma −1 should be m a −1 (milliyears vs. meters per year)*

**Reply:** corrected

*P15, L16–17: This is confusing, since in Section 3 you have argued that the ice mélange quickly adjust its geometry. Why not just use the adaptive approach?*

**Reply:** This experiment corresponds to a mélange with a fixed length which retreats with the glacier. This has been clarified.

*P18, L1: I think you can express the "position-based" calving parameterizations in terms of rates that depend on the thickness gradient. I'm not sure if that is written up anywhere, but my point is that you can probably use this framework for other types of parameterizations than what you have considered here*

**Reply:** Maybe, but we think this goes beyond the scope of this paper.

**Referee 2**

*One thing that is still missing and would be useful is a paragraph (perhaps towards the end) summarizing and discussing the implications of the limitations of this melange parameterization that were introduced by the necessary assumptions and simplification steps.*
*A big one in my view, is the assumption that the melange thickness is the primary control on its strength. See Robel at al for example where he shows that it is the weakening bonds between individual icebergs that determine the melange strength. Also there are works suggesting that there is a limit to the strength the ice melange can exert at the calving front and that is because of buckling.*

**Reply:** Done.

*Finally, while what the authors derive is an upper bound, there are some effects that will make this upper bound actually higher, like ocean currents advecting icebergs away and melting them. So it is not a true upper bound.*

**Reply:** Correct. This has been added to the discussion in section 2.

*Related to the above, I still think it would be insightful to diagnose the melange stress from the numerical model. This would allow an assessment of whether the parametrization causes realistic values of melange-caused backstress. Showing a time series of effective stress at the calving front for the different modeled scenarios would be sufficient and possibly informative.*

**Reply:** The model does not include mélange stress, because the reduction in calving rates is achieved through the assumption that calving rates decrease linearly with mélange thickness and that calving is suppressed once mélange thickness reaches a fraction gamma of the ice thickness. However an equation for the force per unit width as a function of mélange thickness from Amundsen&Burton2018 has been used to estimate the mélange stress.

*I appreciate that the authors included a table of symbols now, still though some of the symbols are missing from the table, e.g. "d".*

**Reply:** d has been replaced in the text

*There are some leftovers from the previous correction where the term "effective melange thickness" was introduced. I think the others decided to go with "melange thickness" so that should be consistently used everywhere. similarly "d" is still present, but undefined.*

**Reply:** This has been corrected in several places

*I think what the authors propose is a parameterization of melange buttressing effects, not a model of melange buttressing. The title and the abstract should reflect that - using the world parameterization would be enough to fix that.*

**Reply:** Done

*P3L5: Observations at... instead of Observations in...*

**Reply:** ?

*P7L5: mélange thinning instead of mélange thickness thinning*

**Reply:** corrected

*P9L10: the instead of die*

**Reply:** corrected

*P15L5: the choice instead of choice*

**Reply:** ?

[revised manuscript text omitted]

---

## Author Response (AR3)

Dear editors,

here are the production files.

Bests,

Anders Levermann